# Methane-cycling microbiomes in soils of the pan-Arctic and their response to permafrost degradation
Haitao Wang [1] ✉, Erik Lindemann [1], Patrick Liebmann [2], Milan Varsadiya[3],
Mette Marianne Svenning[4], Muhammad Waqas[3], Sebastian Petters[1,5], Andreas Richter [6],
Georg Guggenberger [2], Jiri Barta[3] & Tim Urich[1] ✉

The methane-cycling microbiomes play crucial roles in methane dynamics. However, little is known about their distributions on a pan-Arctic scale as well as their responses to the widespread permafrost degradation. Based on 621 datasets of 16S rRNA gene amplicons from intact permafrost soils across the pan-Arctic, we identified only 22 methanogen and 26 methanotroph phylotypes. Their relative abundances varied significantly between sites and soil horizons. Only four methanogen phylotypes were detected at all locations. Remarkably, the permafrost soil methane filter was almost exclusively dominated by some obligate methanotroph (*Methylobacter*-like) phylotypes. However, a case study in Alaska suggests that atmospheric methane oxidizing bacteria (*Methylocapsa*-like phylotypes) dominated methanotrophs in a drier condition after permafrost degradation. These findings point towards a few key microbes particularly relevant for future studies on Arctic methane dynamics in a warming climate and that under future dry conditions, increased atmospheric methane uptake in Arctic upland soils may occur.

Permafrost is primarily distributed across the Arctic region, underlying ~25% of the land area in the Northern Hemisphere and hosting an estimated carbon stock of 1500–1700 Pg[1,2]. Permafrost thaw due to climate change leads to the thickening of the active layer, which releases the frozen carbon and increases the availability and mobility of soil organic carbon (SOC)[3,4]. This ongoing effect accelerates the potential of microbial organic matter decomposition and is thought to increase emissions of greenhouse gases, $CH_4$ and $CO_2$[2,5], which may result in a positive feedback effect into the climate system[1]. Depending on ice richness and soil drainage, permafrost thaw can result in water-saturated soils due to the abrupt collapse of ice wedges[6,7], or drier soils due to better drainage and evapotranspiration[8,9]. While drier and aerated soils may host a burst of $CO_2$ via the microbial degradation of ancient carbon[1,10], anoxic conditions in wetter soils may hamper this process and contribute to SOC accumulation[11]. Although well-drained Arctic soils are well known $CH_4$ sinks[12], anoxia in water-saturated soils favors anaerobic methanogenesis and the expansion of anoxic conditions after permafrost thaw may enhance the proportion of future $CH_4$ emissions[13].

$CH_4$ has ~34 times higher warming potential compared to $CO_2$ on a 100-year timescale[14]. Biogenic $CH_4$ is primarily produced by methanogenic archaea carrying out anaerobic methanogenesis, despite some bacteria and eukaryotes producing $CH_4$ using different mechanisms[15–17]. Anaerobic methanogenic archaea constitute the major archaeal communities in permafrost soils[18,19]. Typical methanogens are a diverse archaeal group occurring within 8 validly described orders, *Methanobacteriales*, *Methanococcales*, *Methanopyrales*, *Methanocellales*, *Methanomicrobiales*, *Methanonatronaechaeales*, *Methanosarciniales* and *Methanomassiliicoccales*[20,21]. Some uncultured lineages, e.g., *Ca.* Bathyarchaeota, *Ca.* Methanofastidiosa and *Ca.* Verstraetearchaeota, are speculated to also carry out methanogenesis based on the coding potential of their metagenome-assembled genomes[21]. The evidence was recently found in *Methanosuratincola* (formerly affiliated with *Ca.* Verstraetearchaeota) within Thermoproteota[22,23], and even in Korarchaeia[24]. Methanogens consist of three main metabolic groups: acetoclastic, hydrogenotrophic and methylotrophic methanogens, utilizing acetate, $H_2/CO_2$ and formate, and methanol/methylamines/

[1]Institute of Microbiology, University of Greifswald, Greifswald, Germany. [2]Institute of Earth System Sciences, Section Soil Science, Leibniz Universität Hannover, Hannover, Germany. [3]Department of Ecosystem Biology, University of South Bohemia, České Budějovice, Czech Republic. [4]Department of Arctic and Marine Biology, The Arctic University of Norway, Tromsø, Norway. [5]The Norwegian College of Fishery Science, The Arctic University of Norway, Tromsø, Norway. [6]Centre for Microbiology and Environmental Systems Science, University of Vienna, Vienna, Austria. ✉e-mail: haitao.wang@uni-greifswald.de; tim.urich@uni-greifswald.de

methyl-sulfides/ethanol as their substrates, respectively[25]. These pathways are the major players in methanogenesis in soils, despite recent findings on two novel methanogenesis pathways namely methoxydotrophy[26] and alkylotrophy[27].

In permafrost environments, 20%-60% of the microbial-source $CH_4$ can be consumed by methanotrophs before emitting to the atmosphere[28,29]. Methanotrophs consist of aerobic methane oxidizing bacteria (MOB) within Alphaproteobacteria, Gammaproteobacteria and Verrucomicrobia, as well as anaerobic bacteria *Methylomirabilales* and anaerobic archaeal methanotroph (ANME) groups within *Methanosarcinales*[25,30]. Aerobic MOB of Alpha- and Gamma-proteobacteria are classified as type II and type I methanotrophs, respectively, and they are the major methane oxidizers occurring in most environments[30]. While these MOB mostly oxidize the microbial-source $CH_4$ in soils and sediments[31], some groups are able to utilize atmospheric $CH_4$[32]. These atmospheric MOB (atmMOB) have a high-affinity particulate methane monooxygenase and can utilize $CH_4$ in a low-$CH_4$-concentration environment, e.g., the atmosphere[33]. They have been detected in many different soils[34–37]; however, so far, only one strain, *Methylocapsa gorgona* MG08, has been isolated which was also detected in Arctic regions[32]. The atmMOB are gaining increasing attentions as they are responsible for the mitigation of atmospheric $CH_4$, which can help to counteract climate change.

The diversity and abundance of methane-cycling microbiomes are expected to vary across the Arctic due to the heterogeneity in the structure and physicochemical conditions of permafrost soil regions[38]. In soils, both methanogens and methanotrophs are important in determining $CH_4$ fluxes[39–41]. It is therefore pivotal to characterize these microbiomes across space and time, for a fundamental understanding of the key players in $CH_4$ dynamics in permafrost soils. However, due to the reduced accessibility of the Arctic permafrost region for sampling, the diversity of these microbiomes and the dominant microbes in these soils are poorly characterized, particularly across horizons and on a broader, pan-Arctic scale.

Arctic wetlands and water-logged soils are known hotspots of $CH_4$ emissions[42,43]. This is due to the wet conditions after permafrost thaw that contribute to the increase of methanogen abundances and the shift in their community compositions[44,45]. A substantial number of studies have particularly addressed the consumption of atmospheric $CH_4$ in well-drained Arctic soils[12,46–50], but surprisingly little attention has been paid to the microbiomes driving this phenomenon. Only a few studies mentioned atmMOB as the main drivers of $CH_4$ consumption in Arctic soils[37,51,52]. Moreover, studies comparing the abundance and composition of methane-cycling microbiomes across soil moisture gradients following permafrost thaw remain scarce.

This study aims to reveal the pan-Arctic distributions of methanogens and methanotrophs in intact permafrost soils and their future development in response to different water conditions after permafrost thaw. We analyzed the microbiomes of intact permafrost soil samples from eight locations across the pan-Arctic over a course of eight years (Fig. 1a) from different horizons (Fig. 1b). The relative abundance and distribution patterns of phylotypes associated with methanogens and methanotrophs across space and horizons are shown. To understand the impact of permafrost degradation on those methane-cycling microbiomes in times of climate change, we selected three hydrologically different degraded permafrost sites and their corresponding intact site in Fairbanks, Alaska. The relative abundance and community composition of methanogens and methanotrophs were compared among the degraded sites and between the degraded sites and the intact site.

## Materials and methods
### Studying sites and soil sampling
Over the course of 8 years, soils samples were taken from 8 different intact Arctic sites located in the continuous permafrost domain (Fig. 1a), including Herschel Island (2016) and Beaufort Coast (2016) in Canada, Disko Island (2017) and Zackenberg (2010) in Greenland, and Logata (2011), AriMas (2011), Tazovskiy (2012) and Cherskiy (2010) in Siberia, Russia. For each sampling, soil samples were taken from different horizons, including organic layer, topsoil, subsoil, cryoturbated organic matter (cryoOM) and permafrost (Fig. 1b). The sampling campaigns were conducted in summer (July or August) at the peak of the growing season within different projects. The detailed descriptions of these sites and sampling protocols are available

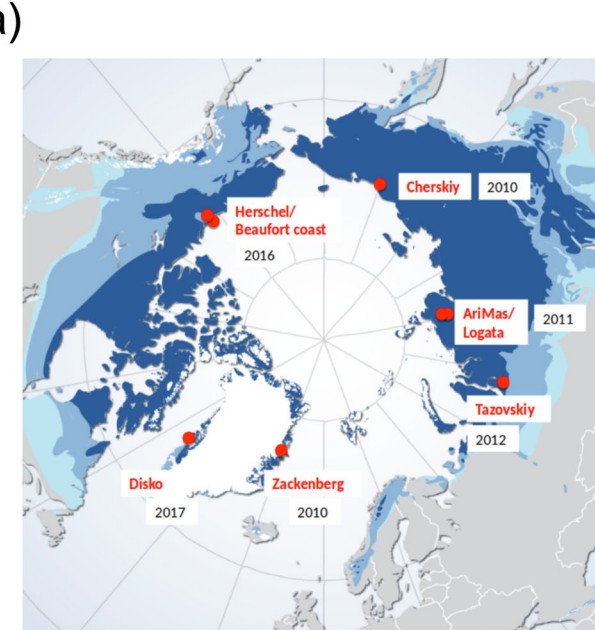

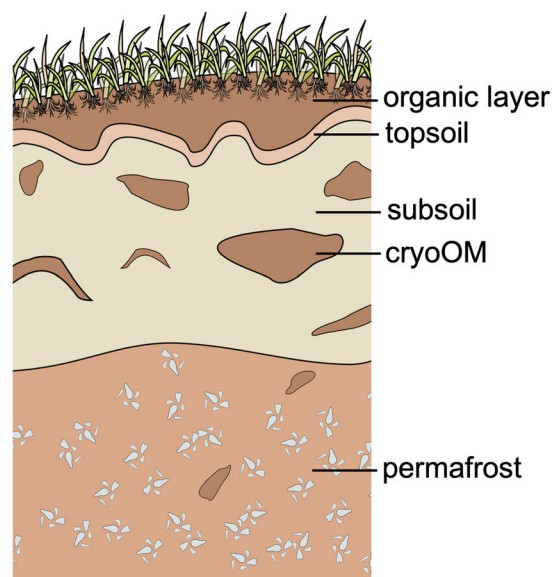

**Fig. 1 | Sampling sites and horizons in the pan-Arctic. a** Sampling locations including (from west to east) Herschel Island and Beaufort coast (Canada), Disko Island and Zackenberg (Greenland), as well as Tazovskiy Peninsula, AriMas, Logata and Cherskiy (Russia). **b** Schematic diagram of soil horizons including organic layer, topsoil, subsoil, subsoil cryoturbated organic matter (cryoOM) and permafrost.

The base map in (**a**) is from https://www.grida.no/resources/5234 (Hugo Ahlenius and UNEP/GRID-Arendal, 2016). Dark blue refers to continuous permafrost >90% area coverage; medium blue refers to discontinuous/sporadic 10–90% coverage; light blue refers to isolated patches; white over land refers to no permafrost.

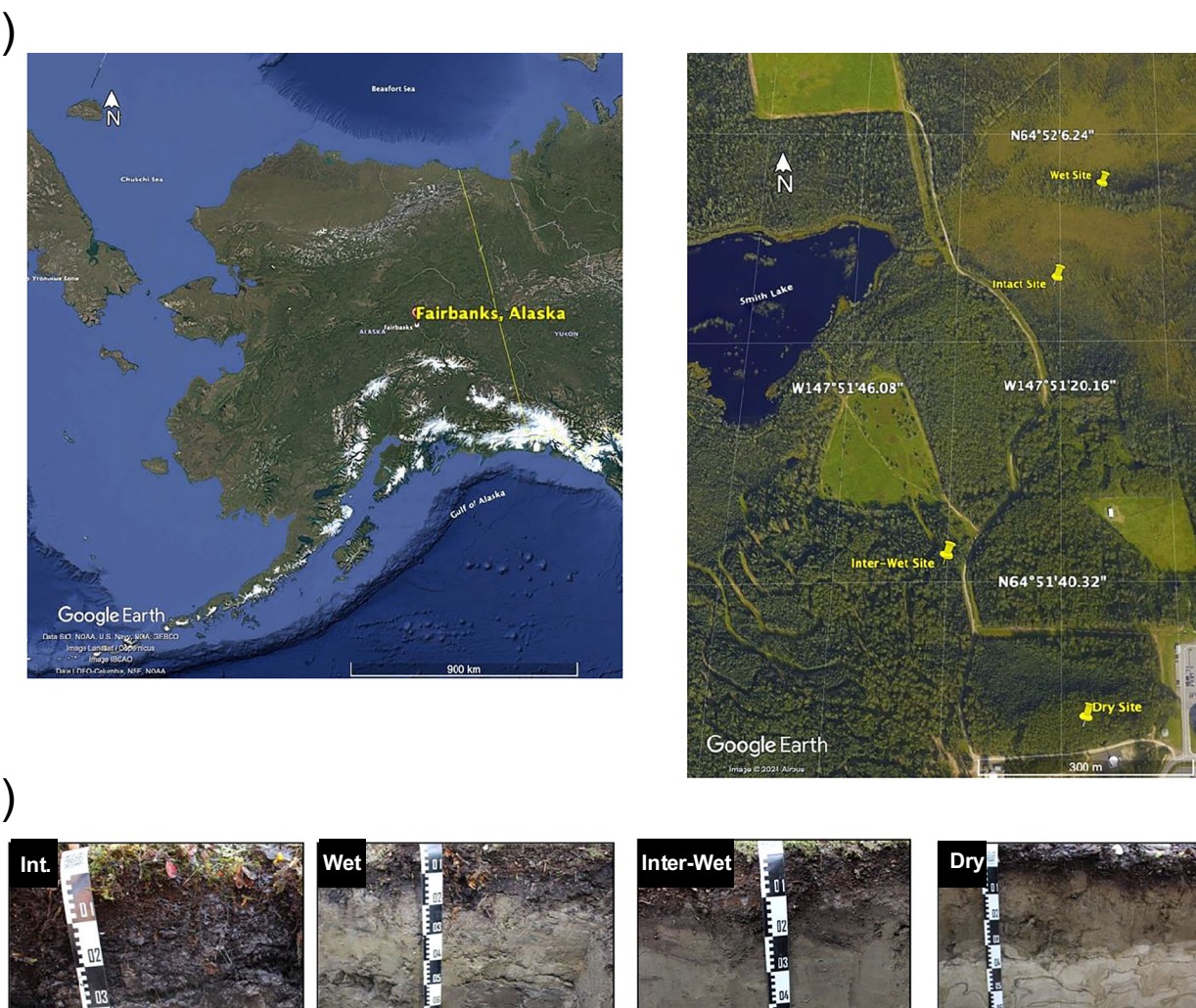

**Fig. 2 | Sampling sites and soil profiles in Alaska. a** Sampling locations in Fairbanks, Alaska. **b** Soil profiles of the four sites with different water conditions. Int. Intact, Inter-Wet Intermediate-Wet. This figure is modified from a previous study[56].

in previous studies[19,53–55]. The number of samples taken at each layer in each site is shown in Supplementary Fig. S2 and Supplementary Fig. S3. In total, 621 samples were collected.

A ninth intact permafrost site is located in the city of Fairbanks, in the Interior Alaska, USA (Fig. 2a). This Intact permafrost site (64° 51′ 56.1″ N, 147° 51′ 18.9″ W) is close to Smith Lake with a shallow permafrost table and no major degradation features (Fig. 2b), like other intact sites from the pan-Arctic (Fig. 1a). To study the impact of different water conditions on microbiomes after permafrost thaw, two degraded permafrost sites in the vicinity of the Intact site were selected. The Wet degraded permafrost site (64° 52′ 02.4″ N, 147° 51′ 17.3″ W) is located on a low hill at the bottom of the depression with degraded permafrost soils and high-water contents (Fig. 2b). The Dry degraded permafrost site (64° 51′ 33.5″ N, 147° 51′ 18.0″ W) is at the shoulder of a north-facing slope with well-aerated and degraded permafrost soils and low water contents (Fig. 2b). A detailed description of these three sites is available in a previous study[56]. Additionally, a third degraded permafrost site was investigated on a mid-slope position (64° 51′ 41″ N, 147° 51′ 33″ W) with soil development and moisture conditions being in between those of Dry and Wet sites (Fig. 2b), in the following referred to as Intermediate-Wet site.

Soil samples in Alaska were taken in late August to early September in 2019 and 2021. In 2019, each three soil pits (150 × 100 cm) were dug with a depth of either ~100 cm for degraded (Dry and Intermediate-Wet) sites or until reaching the permafrost surface for the non-degraded (Intact) site. In 2021, an additional site with a higher moisture content (Wet site) was sampled. Only one pit was dug at each site and the same approach was used for sampling the profile as in 2019. In compensation, we additionally sampled soils from 6 satellite pits at the depths of top- and sub-soil layers (~5 cm or ~50 cm, respectively) to account for the large, small-scale heterogeneity of the soils in this area. The number of samples taken from each different depth is shown in Supplementary Fig. S4.

### Soil physicochemical properties

The soil water content and pH were measured for samples from the 8 pan-Arctic sites (Fig. 1a). Water content was measured as water to fresh soil weight ratio by drying the soil at 105 °C until weight stabilized. The pH values were measured using a soil-water-suspension with a fixed 5:2 ratio (water to soil). The water content of Disko samples and the pH of Zackenberg samples were not available due to different managements and measurements of the involving projects.

Soil samples taken in Alaska were analyzed on a variety of soil physicochemical parameters, including soil moisture, pH, soil organic carbon (SOC), total nitrogen (TN), base saturation (BS), dissolved organic carbon (DOC) and dissolved total nitrogen (DTN), micro- and macro-aggregate (MiA and MaA) proportions, and organic carbon and nitrogen in MiA and MaA. Data were adapted from a previous study[56].

## Microbiome analysis

Soil samples from the 8 pan-Arctic sites (Fig. 1a) were processed with DNA extraction and 16S rRNA gene amplicon sequencing using the primer pair 515F/806R. Illumina sequencing with pair-end 100 bp was performed with Cherskiy and Zackenberg samples collected in 2010, while the other samples were sequenced with a later pair-end 250 bp Illumina platform. The sequencing data of the pan-Arctic samples were generated and published by previous studies[19,53–55]. To make all sites comparable, forward sequences were all trimmed to 100 bp and then used for downstream analysis. The UNOISE algorithm[57] was used to denoise and error-correct the sequencing data, and zero-radius Operational Taxonomic Units (ZOTUs) were generated. The taxonomy of each ZOTU was assigned against the SILVA 138.2 database using a Naive Bayesian Classifier algorithm implemented in *dada2* pipeline[58] in R v3.6.3[59]. Based on 621 analyzed datasets, a subset of ZOTUs associated with methanogens and methanotrophs (Supplementary Table S1) were used for downstream analyses. The core members that existed in all sites except for Tazovskiy (due to absence or low abundances of both methanogens and methanotrophs) were identified. The representative sequences of associated ZOTUs were further used as enquiry for NCBI BLASTN against the 16S rRNA gene sequence (bacteria and archaea) database to verify the taxonomy identification (Supplementary Data 1).

Alaska soil samples (Fig. 2a) taken in 2019 were transported on ice and kept at −20 °C before the DNA extraction. DNA was co-extracted with the RNA using the RNeasy PowerSoil total RNA kit and the RNeasy PowerSoil DNA elution kit (Qiagen, Hilden, Germany) from ~2 g of homogenized soil samples. In 2021, ~3 g of each soil was mixed with 2 volumes of LifeGuard Soil Preservation Solution (Qiagen, Hilden, Germany) directly in the field. All treated soils were kept at 4 °C before the DNA extraction. DNA was also co-extracted with the RNA using the same kits. The LifeGuard solution was removed by centrifugation and ~2 g of the treated soil was used for the extraction. The extracted DNA samples were used for 16S rRNA amplicon sequencing using Illumina platform (pair-end 250 bp) using the primer pair 515F/806R. The sequencing data of the Alaska samples was generated for this study and has not been published elsewhere. These data were analyzed separately and the phylotypes of methanogens and methanotrophs were inferred as amplicon-sequencing-variants (ASVs). However, ASVs and ZOTUs both provide species-level resolved phylotypes[57,58] and are thus comparable with each other.

The *dada2* pipeline was used to process the Alaska data. Sequences failing to meet the filter criteria (maxEE = 2, truncQ = 2, maxN = 0) were removed. Those filtered sequences were de-replicated, the ASVs were deduced, and the paired-end sequences were merged. Afterwards the chimeric sequences were removed. The sequence of each ASV was assigned to taxonomy using the SILVA 138.2 database as described above. ASVs associated with methanogens and methanotrophs were identified with the same criteria (Supplementary Table S1). Those ASV sequences were also used as query to verify the taxonomy identification on NCBI as described above (Supplementary Data 1). The *Beijerinckiaceae* ASVs were additionally used as query against 16S rRNA gene of *M. gorgona* MG08 with a local BLASTN to verify their relationships (Supplementary Data 1). To link the methanotroph ASVs in Alaska to ZOTUs in the pan-Arctic, a local BLASTN was run using ASVs as the enquiry and ZOTUs as the database. Only ASVs showing a 100% identity to a ZOTU are considered as the potential same phylotype.

## Statistics

The downstream analyses were done with R v3.6.3[59]. The normality of abundance data distributions was checked before linear regressions or t tests

using the Shapiro-Wilk test. If not normally distributed, the data were $log_{10}$ transformed. A minimum value was added to avoid zeros when necessary, before the transformation. The coefficients and significance level of the correlations were determined by Pearson's correlation analysis using *vegan* v2.5.6 package[60]. The significant correlations were further confirmed by non-parametric Spearman's correlation analysis using *vegan*. Pairwise *t* tests were performed to compare the means of methanogen and methanotroph relative abundances between each two sites or horizons. Spearman's correlation was also used to check the correlations between methanogen and methanotroph abundances and environmental parameters for Alaska samples. For all multiple comparisons, *P* values were adjusted by the false discovery rate (FDR) method. All of the plots were generated using *ggplot2* v3.3.3 package[61].

## Reporting summary

Further information on research design is available in the Nature Portfolio Reporting Summary linked to this article.

## Results and discussion

### Functional guild abundances across the pan-Arctic and soil horizons

Methanogen relative abundance (in total microbiome) varied strongly between locations, from the highest found in Herschel Island (~0.5%) and Zackenberg (~0.75%) to almost undetected in Disko Island and Tazovskiy (Fig. 3a). The methanogen distribution along the vertical profiles was often location-specific, sometimes increasing with depth, and sometimes having the highest abundance in the organic layers (Fig.3c and Supplementary Fig. S2). In mineral top- and subsoils, methanogens were lowly abundant likely due to the lack of substrates. Nevertheless, their relative abundance and soil water content were significantly positively correlated in organic layer, cryoOM and permafrost (Supplementary Fig. S1a) in which the methanogens were also abundant (Fig. 3c). This indicates that the hydrologic status is one key driving factor for methanogen abundance by limiting oxygen diffusion into the soil profile in layers where they habitat. pH only showed a significant and negative correlation with methanogen relative abundances in the permafrost layer (Supplementary Fig. S1c).

Methanotroph relative abundance (in total microbiome) followed the methanogen pattern, with the highest (~0.32%) and lowest (almost undetected) abundances found in Zackenberg and Tazovskiy, respectively (Fig. 3b). Along the soil profiles, methanotrophs were less abundant in the organic layer and topsoil compared to the other deeper layers (Fig. 3d), but this pattern again varied between sites (Supplementary Fig. S3). Compared to methanogens, water content only showed a weak but still significant correlation with methanotroph abundance in the organic layer (Supplementary Fig. S1b). Like methanogens, methanotroph abundance was negatively and significantly correlated with pH only in the permafrost layer (Supplementary Fig. S1d). Overall, methanotroph abundances showed a strong and positive correlation with methanogen abundances in all layers except permafrost (Supplementary Fig. S1e). This suggests that these methanotrophs in the active layers likely rely on the $CH_4$ produced by the methanogens. While methanogen abundances could be mainly driven by abiotic factors such as water (Supplementary Fig. S1a), methanotroph abundances could be driven by the resource availability in pan-Arctic permafrost affected soils.

### Functional guild compositions across the pan-Arctic and soil horizons

Methanogens conducting hydrogenotrophic methanogenesis dominated the methane producing community (Fig. 4a). We considered phylotypes found in all locations (excluding Tazovskiy) as members of the pan-arctic core methanogenic microbiome. One member was associated with the uncultured candidatus family *Methanoflorentaceae* (former Rice-cluster-II) which had been found widespread in thawing permafrost[44,62]. The other phylotype was associated with the $H_2$-dependent methylotrophic order *Methanomassiliicoccales*. Another one was closely related with

**Fig. 3 | Relative abundances of methanogens and methanotrophs in the pan-Arctic. a, b** Relative abundances of methanogens (**a**) and methanotrophs (**b**) in different sites. **c, d** Relative abundances of methanogens (**c**) and methanotrophs (**d**) in different horizons. Abundances are shown as mean + standard error. The bars with no common letter are significantly different ($P < 0.05$) characterized by pairwise $t$ tests with $P$ values adjusted by the false discovery rate method.

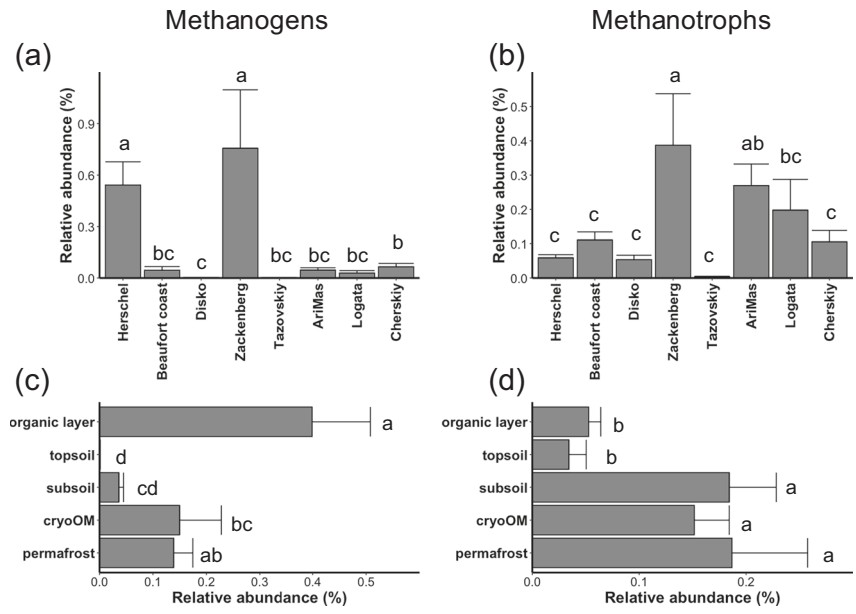

*Methanosarcina* which can use multiple methanogenesis pathways. The most abundant phylotype, however, was closely related to *Methanobacterium* which can perform methanogenesis from both $H_2/CO_2$ and methanol/$H_2$ pathways.

The pan-Arctic methanotrophs were dominated by type I MOB (Fig. 4b). Only two phylotypes of the *Ca.* Methanoperedens clade (formerly ANME-2d) were found and one was notably abundant across many locations, especially in the subsoil (Fig. 4b), suggesting a potential vital role of anaerobic methane oxidation in mitigating methane emissions in anoxic deeper layers of permafrost soils. Similarly, only two type II MOB phylotypes related to *Methylocella* were found and one was found in all the sites (Fig. 4b). They are known as facultative methanotrophs that can grow on low-molecular-weight organic compounds[30]. Among the detected type I MOB phylotypes, nine were related to the obligate methanotroph *Methylobacter* and seven were found as the core methanotrophs (Fig. 4b). Taken together, these *Methylobacter*-like phylotypes accounted for in average 77% of the total methanotrophic community. This proportion varied across studied locations, ranging from min. 28% (Tazovskiy, Russia) to max. 98% (Disko, Greenland). This finding is remarkable, since it suggests that the microbial $CH_4$ filter in permafrost-affected soils across the Arctic is of a strikingly low diversity and dominated by one single genus. Further, NCBI BLASTN results suggested that these *Methylobacter*-like phylotypes were all closely related to *M. tundripaludum* SV96 (95–100% identity, Supplementary Data 1) which was first isolated from Svalbard peatlands[63]. This hints that the pan-Arctic microbial $CH_4$ filter might even be dominated by one single species, although the species-level identity is not robust based a short fragment (~250 bp) of 16S rRNA gene. *M. tundripaludum* was also found as dominant methanotrophs in other locations on Svalbard[64] and in the Lena Delta in Siberia[65], which supports our finding. Moreover, *M. tundripaludum* was found as an active methane oxidizer in high Arctic wetlands using stable isotope probing[66]. A recent study showed physiological adaptions of *M. tundripaludum* to varying temperatures by adjusting their central metabolism, protein biosynthesis, cell walls and storage[67]. This might explain the prevalence and dominance of *M. tundripaludum* across heterogeneous Arctic regions.

**Responses to different water status after permafrost degradation – case study in Alaska**

The relative abundance of methanogens in the Intact site was <0.01% in both 2019 and 2021 (Supplementary Fig. S4a), which is among the lowest relative abundances detected in the other intact sites across the pan-Arctic (Fig. 3a).

Accordingly, only one phylotype related with *Methanobacterium* was found in the Intact site in 2021 (Fig. 5a). However, 4 phylotypes of the order Methanomassiliicoccales were mainly found under Dry and Intermediate-Wet conditions (Fig. 5a), but still only accounting for up to 0.03% of the total microbiome (Supplementary Fig. S4a). They were only detected in the deeper soil layers (>30 cm below the surface) (Fig. 5a). The low abundance in the Intact site of Alaska might be due to the shallower sampling depth as more methanogens were found in the layers lower than 40–60 cm in the other sites (Fig. 5a). The deeper layers in the Intact site were frozen permafrost in which microbes should not be active despite their existence.

The methanotrophs were less diverse in the Intact site compared to the other pan-Arctic sites (Figs. 4b and 5b). Only *Methylobacter*-like phylotypes were detected in both 2019 and 2021. Their relative abundance in the Intact site is comparable to that in Herschel Island and Disko Island (Fig. 3b and Supplementary Fig. S4b). In contrast, the methanotrophic communities in Dry and Wet sites were dominated only by *Methylocapsa*-like and *Methylobacter*-like phylotypes, respectively (Fig. 5b). In the Intermediate-Wet site, both groups were found, with additionally *Ca.* Methylomirabilis phylotypes detected (Fig. 5b). Compared to the Intact site, the relative abundance of methanotroph was higher in the Dry and Intermediate-Wet sites and similar in the Wet site (Supplementary Fig. S4b).

All type I MOB were only found in depth >30 cm below the surface where methanogens were detected, while the type II MOB in the Dry site were mostly found in upper soil layers, with a few found in the deeper layers (Fig. 5a, b). Similar to the pan-Arctic sites, *Methylobacter*-like phylotypes dominated the MOB community in the Intact and Wet sites (Fig. 5b). This is astonishing as their dominance is consistent across the pan-Arctic permafrost affected soils, even including the Wet degraded permafrost site. We ran a local BLASTN to link Alaska ASVs to the pan-Arctic ZOTUs. Most of the type I MOB ASVs could be linked to a ZOTU with 100% identity (Supplementary Table S2), suggesting that most of these detected Alaska *Methylobacter* were likely identical to those found at the other pan-Arctic sites.

The distribution pattern of methanotrophs in different Alaska sites is likely due to the different water conditions. In the Dry site, the *Methylocapsa*-like phylotypes are closely associated with *M. palsarum* and *M. gorgona* (>97% identity, Supplementary Data 1). It is known that *Methylocapsa* hosts atmMOB and that *M. gorgona* MG08 is so far the only isolated atmMOB[32]. Due to their high-affinity to $CH_4$, these MOB can utilize $CH_4$ at very low $CH_4$ concentrations, e.g., the atmospheric level[33]. This explains why the Dry site hosted only *Methylocapsa* MOB. In the surface layer, dry

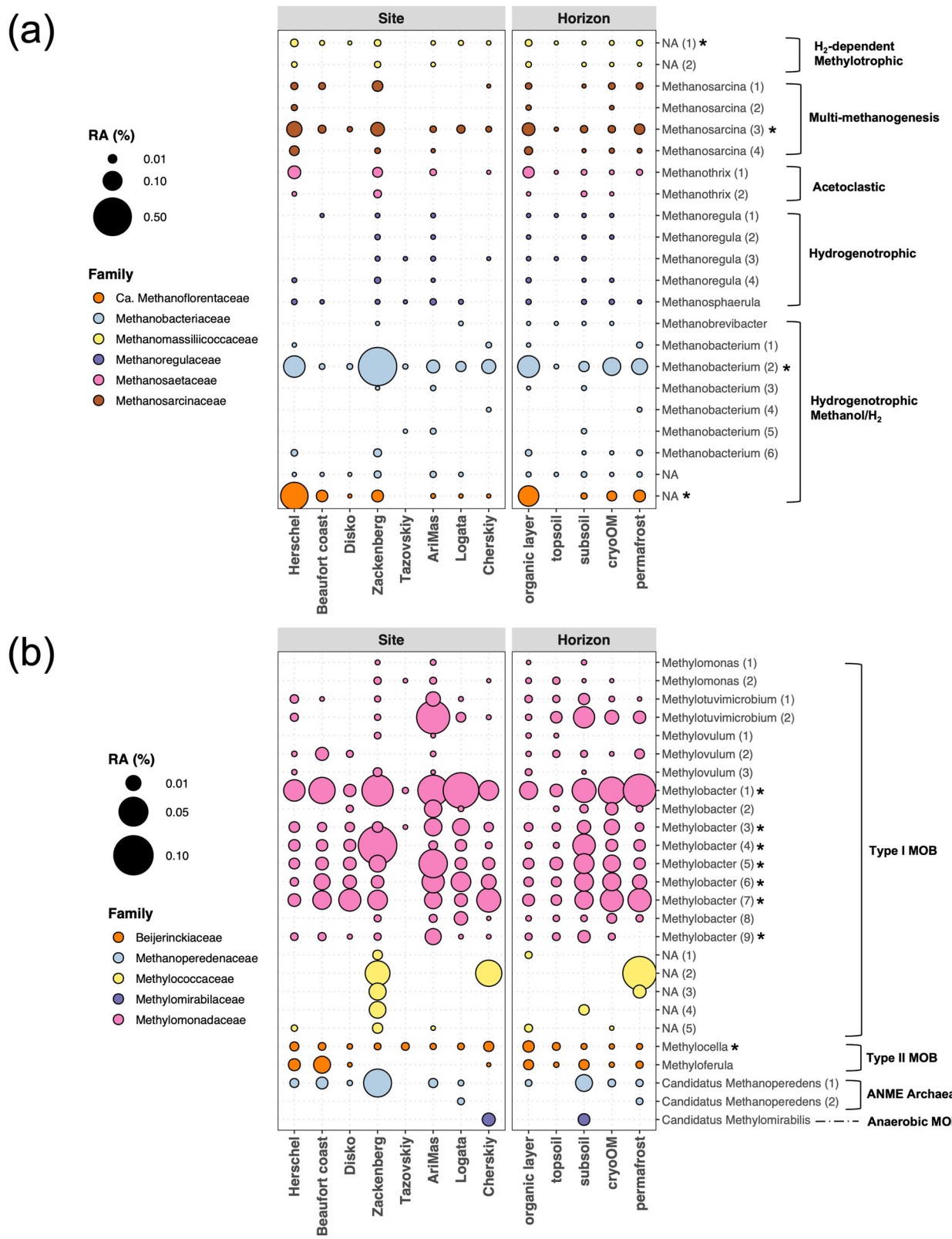

**Fig. 4 | Distributions of methanogen and methanotroph phylotypes in different sites and horizons in the pan-Arctic. a, b** The relative abundance of each methanogen (**a**) and methanotroph (**b**) phylotype. Numbers in brackets indicate different phylotypes that were assigned to that genus as shown in Supplementary Data 1. RA relative abundance, NA not assigned at genus level, MOB methane oxidizing bacteria, ANME anaerobic methanotroph, * core methanogen or methanotroph members that were found in all sites (excluding Tazovskiy).

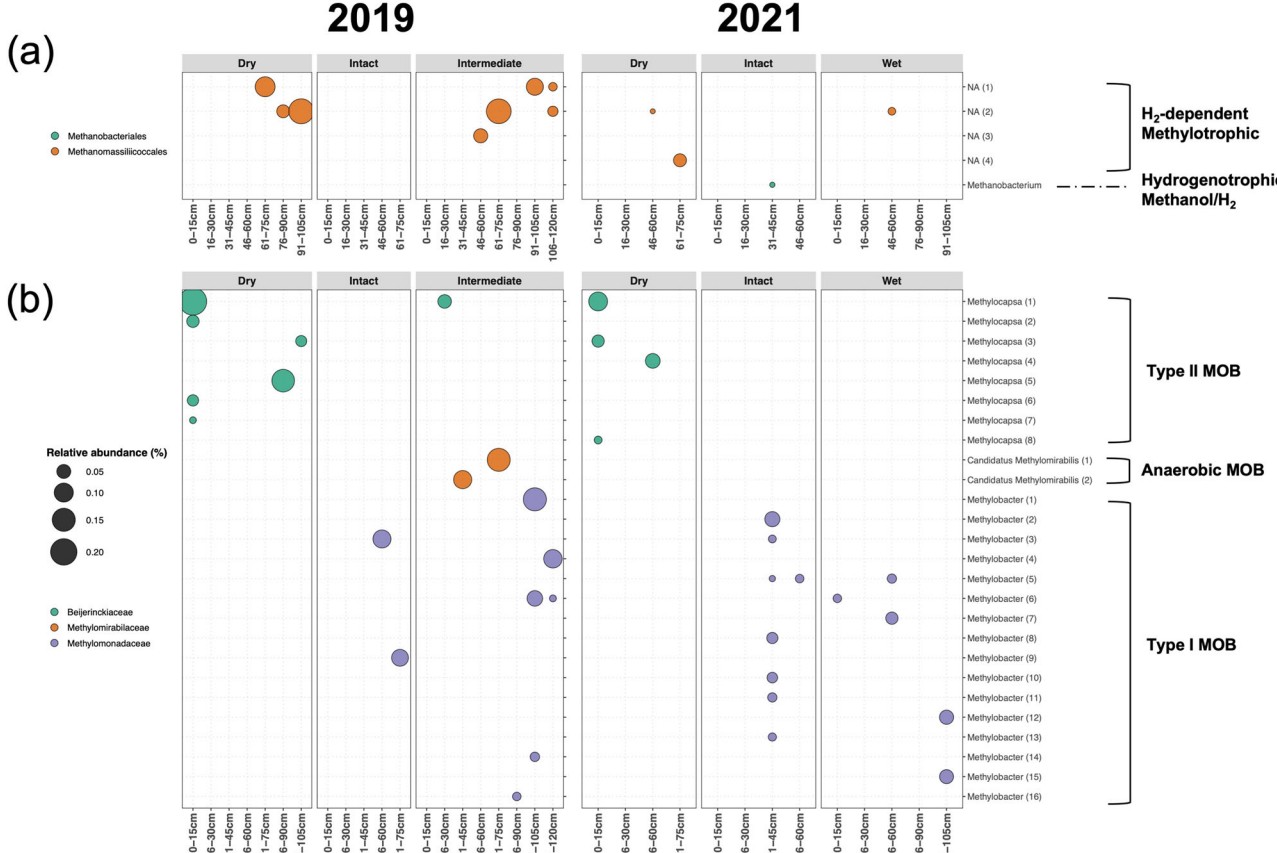

**Fig. 5 | Distributions of methanogen and methanotroph phylotypes in different sites and depths in the Alaska study. a, b** The relative abundance of each methanogen (**a**) and methanotroph (**b**) phylotype in 2019 and 2021. Numbers in brackets indicate different phylotypes that were assigned to that genus as shown in Supplementary Data 1. RA relative abundance, NA not assigned at genus level, MOB methane oxidizing bacteria, ANME anaerobic methanotroph.

conditions create aerobic environments which attenuate anaerobic methanogenesis, and the lack of microbial $CH_4$ source further hampers the "conventional" but not the atmMOB. In deep layers of the Dry site, some methanogens were still found but they were likely not active since they were almost absent in 2021 (Fig. 5a). The presence of *Methylocapsa* MOB in deeper layers might be because they are facultative methanotrophs which could use alternative substrates other than $CH_4$[32].

In contrast, the Intact and Wet sites only hosted "conventional" type I MOB in deeper layers although almost no methanogens were detected. In the Intermediate-Wet site, the dominant MOB switched from *Methylocapsa* in top layers to *Ca.* Methylomirabilis in middle layers then to *Methylobacter* in deep layers (Fig. 5b). The anaerobic and type I MOB found in the middle and deeper layers might be supported by the methanogens living in these layers (Fig. 5a). However, this relationship could not be identified in the other sites, suggesting either that these MOB were inactive or that other $CH_4$ sources (e.g., plant-associated or abiotic sources) supported them in these sites.

### Factors driving methane-cycling microbiome changes in intact and degraded sites

Methanogens and type I MOB showed congruent responses to the soil physiochemical properties (Fig. 6). They both showed a positive correlation with depth and pH, but a negative correlation with SOC, TN, C/N and temperature in 2019 (Fig. 6a). In contrary, type II MOB positively correlated with SOC, TN, C/N and temperature, but negatively with depth, moisture and pH (Fig. 6a). In 2021, most correlations were insignificant (Fig. 6b). However, depth, moisture, pH, SOC, TN and C/N showed a same relationship with methanogens, type I and type II MOB, respectively, as compared to 2019 (Fig. 6a, b).

The primary factor determining $CH_4$ uptake in Arctic soils is moisture, with lower moisture leading to a higher uptake[12,49]. In a temporal scenario, temperature is a more important factor driving seasonal dynamics of $CH_4$ fluxes[47,50]. Only a few studies pointed out the importance of biotic factors[12,46,68], but without valid support of microbiome data. However, these abiotic factors mainly impact the microbial drivers, i.e., atmMOB conducting the primary biogenic mitigation of atmospheric $CH_4$. Here, we show that under well-drained conditions after permafrost thaw, the methanotroph communities comprised exclusively atmMOB associated phylotypes, supporting the observed $CH_4$ uptake and interpretation on atmMOB in dry Arctic landscapes by previous studies[12,48,68]. Our data further indicated that moisture and temperature are important drivers for methanotrophs as well because of their significant correlations (Fig. 6a). The transition from type I in the Wet site to a mixture of both types in the Intermediate-Wet site, and to type II in the Dry site also suggested that the water condition associated changes led to the shifts in methanotroph community compositions that may eventually determine the $CH_4$ uptake.

The correlations further suggested that higher SOC content may favor the proportion of type II MOB (Fig. 6). This is because higher SOC input can provide alternative carbon substrates that enhance their growth as many type II MOB are facultative mixotrophs that can grow on low-molecular-weight organic compounds[30]. This is in line with a recent study that found an identical relationship of type II MOB with SOC in paddy soils[69]. The negative and positive impact of pH on type II and type I MOB, respectively (Fig. 6) suggested that pH may play an important role in the niche differentiation of methanotrophs. This was proved by a previous study showing that type II MOB were responsible for consuming $CH_4$ at a relatively low pH, while type I MOB dominated this process at a high pH[70]. The first

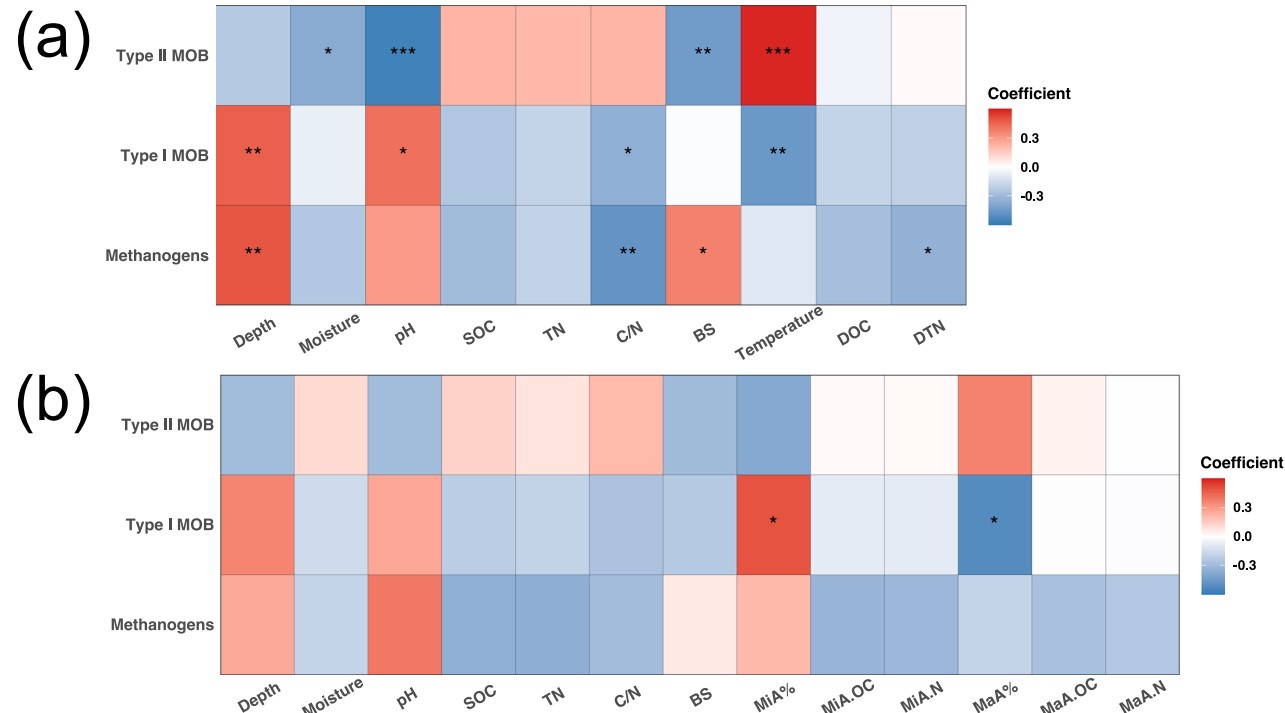

**Fig. 6 | Spearman's correlations between relative abundance of methanogens and methanotrophs (type I and type II MOB) and soil physiochemical properties in the Alaska study. a, b** Heatmap showing correlations in 2019 (**a**) and 2021 (**b**). Asterisks indicate adjusted *P* values, \*\**P* < 0.05, \*\**P* < 0.01, \*\*\**P* < 0.001. SOC soil organic carbon. TN total nitrogen, DTN dissolved total nitrogen, BS base saturation, MiA micro-aggregation, MaA macro-aggregation.

molecular evidence of active atmMOB was also found in acidic mineral Cryosols[37].

While other studies showed that abiotic biogeochemical factors, e.g., moisture, temperature and pH, are important drivers of $CH_4$ balance in Arctic regions, our data suggested that these abiotic factors might be primarily driving methane-cycling microbiomes which mediate $CH_4$ productions and consumptions. Especially the type II MOB which are responsible for consuming atmospheric $CH_4$ reacted similarly to the changes in these factors as observed with $CH_4$ fluxes in previous studies[12,47,49,50,70]. Therefore, both abiotic and biotic factors (methanogen and methanotroph abundances and compositions) should be considered in future modeling for predicting Arctic $CH_4$ dynamics. More investigations of methanotroph communities in the well drained soils from other Arctic regions are needed to further understand their $CH_4$ sink capacity, particularly in times of climate change when more dry conditions are expected.

**The methane-cycling microbiome in Arctic permafrost: unlocking future climate scenarios**

Our data suggested that both dry and wet conditions after permafrost thaw had an impact on the methanogen and methanotroph communities. There was a slight increase of methanogen relative abundance and the number of their phylotypes in both Dry and Wet or Intermediate-Wet sites (Fig. 5a). However, due to their low diversity and low proportions in microbiome in all sites (Fig. 5a and Supplementary Fig. S4a), it was unclear whether permafrost thaw resulted in effective changes in methanogen communities. On the other hand, the methanotrophs showed a much bigger diversity and a completely contrasting pattern of their community compositions between the Wet and Dry sites. The single dominant group switched completely from *Methylobacter* in the Wet to *Methylocapsa* in the Dry site (Fig. 5b). The latter is known to be closely related with atmMOB, suggesting that drier conditions after permafrost degradation might even result in a $CH_4$ sink as these MOB could consume $CH_4$ from the atmosphere. While the status of being wet or dry after permafrost thaw depends on native factors such as vegetation, topography, ice storage and substrate[71], many thawed areas will

likely experience drier conditions as a result of higher temperatures and increased evaporation in a warming climate[72]. Due to global warming, extreme weather events, such as summer drought, are happening more frequently than before[73], which may even boost these processes. Although investigations from other Arctic regions are needed for validation, our data suggest that atmMOB in drained landscapes after permafrost thaw might play a more important role in $CH_4$ dynamics in a future warmer climate.

**Conclusion**

This study shows the distribution of methane-cycling microbiomes including both methanogens and methanotrophs across pan-Arctic regions and horizons in permafrost-affected soils. Methanogens were more abundant in organic and deep layers with high water contents. Methanotrophs were also more abundant in the rather oxygen-limited deeper layers, pointing to their microaerophilic lifestyle and their dependence on methanogens for the $CH_4$ source. However, these patterns and their relative abundances also varied across locations. Four methanogens with a diverse set of methanogenesis pathways indicate a certain flexibility for different methanogenesis substrates and methanogenic conditions in permafrost. Most strikingly, methanotroph phylotypes closely related with *Methylobacter* dominated this functional guild in the pan-Arctic microbiome as well as in the wet degraded permafrost in Alaska. This indicates that the major microbial $CH_4$ filter in Arctic soils comprise of a single group, the obligate methanotrophs within *Methylobacter*. However, under drier conditions, this group could be replaced by atmMOB, which can result in more $CH_4$ uptake in Arctic regions. These findings are crucial for estimating the future methane filter functioning and magnitude in Arctic tundra and taiga soils after permafrost degradation.

**Data availability**

Sequencing data from Herschel Island, Beaufort Coast, Zackenberg, Tazovskiy, AriMas, Logata and Cherskiy are available from Zenodo with https://doi.org/10.5281/zenodo.16993571. Sequencing data from Disko Island and Alaska are available from European Nucleotide Archive of

European Molecular Biology Laboratory (accession number PRJEB72670 for Disko Island; PRJEB83087 for Alaska).

## Code availability

All R codes and associated data files are available at: https://github.com/haitaowang1/Pan-Arctic_methane_cycling_microbiomes.

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

## Acknowledgements
This study was supported by German Research Foundation (DFG) project GU 406/35-1 and UR 198/4-1, and the Czech Science Foundation (GACR) project 20-21259J. This research was also funded, in part, by the Austrian Science Fund (FWF), grant https://doi.org/10.55776/COE7. The authors thank Marc Piecha for his help with the conceptual figure.

## Author contributions
Haitao Wang: Conceptualization; Investigation; Formal Analysis; Visualization; Writing - Original Draft Preparation. Erik Lindemann: Data Curation; Investigation; Formal Analysis; Visualization. Patrick Liebmann: Investigation; Writing - Review & Editing. Milan Varsadiya: Investigation; Writing - Review & Editing. Mette Marianne Svenning: Writing - Review & Editing. Muhammad Waqas: Writing - Review & Editing. Sebastian Petters: Investigation; Writing - Review & Editing. Andreas Richter: Methodology; Writing - Review & Editing. Georg Guggenberger: Resources; Methodology; Writing - Review & Editing. Jiri Barta: Methodology; Resources; Formal Analysis; Investigation; Writing - Review & Editing. Tim Urich: Validation; Project administration; Funding Acquisition; Supervision; Writing - Review & Editing.

## Funding

## Competing interests
The authors declare no competing interests.
