## [Transparent Peer Review file · Communications Earth & Environment]

Methane-cycling microbiomes in soils of the pan-Arctic and their response to permafrost degradation

Corresponding Author: Dr Haitao Wang

Version 0:

Decision Letter:

Dear Dr Wang,

Your manuscript titled "The methane-cycling microbiome in intact and degraded permafrost soils of the pan-Arctic" has now been seen by 3 reviewers, whose comments are appended below. You will see that they find your work of some potential interest. However, they have raised quite substantial concerns that must be addressed. In light of these comments, we cannot accept the manuscript for publication, but would be interested in considering a revised version that fully addresses these serious concerns.

We hope you will find the reviewers' comments useful as you decide how to proceed. **Additionally, we want you to work on the following editorial thresholds for further consideration in Communications Earth & Environment:**

- * Provide sufficiently new evidence into intact and degraded permafrost soil of pan-Arctic region.
- * Present a detailed and clear discussion on methane oxidation and SOC formation as outlined by Reviewer #1.
- * Fully describe the robustness and provide clarity about the 16S rRNA approach in the methodology

Should additional work allow you to address these criticisms, we would be happy to look at a substantially revised manuscript. If you choose to take up this option, please either highlight all changes in the manuscript text file, or provide a list of the changes to the manuscript with your responses to the reviewers.

When resubmitting, please provide a point-by-point response to the reviewers' comments. Please submit your responses as a separate file, distinct from your cover letter where you can add responses to the Editors' comments that you do not want to be made available to the reviewers. Word files are preferred. We recommend that any figures, tables or graphs that are included in the response to reviewers are also included in the main article or Supplementary Information.

If the revision process takes significantly longer than three months, we will be happy to reconsider your paper at a later date, as long as nothing similar has been accepted for publication at Communications Earth & Environment or published elsewhere in the meantime.

Please use the following link to submit your revised manuscript, point-by-point response to the reviewers' comments with a list of your changes to the manuscript text (which should be in a separate document to any cover letter), a tracked-changes version of the manuscript (as a PDF file) and any completed checklist:

Link Redacted

** This url links to your confidential home page and associated information about manuscripts you may have

submitted or be reviewing for us. If you wish to forward this email to co-authors, please delete the link to your homepage first **

Please do not hesitate to contact us if you have any questions or would like to discuss the required revisions further. Thank you for the opportunity to review your work.

Best regards,

Kate Buckeridge, PhD
Editorial Board Member
Communications Earth & Environment
orcid.org/0000-0002-3267-4216

Somaparna Ghosh, PhD
Associate Editor- Communications Earth & Environment
Consulting Editor - Communications Sustainability

EDITORIAL POLICIES AND FORMAT

If you decide to resubmit your paper, please ensure that your manuscript complies with our editorial policies and complete and upload the checklist below as a Related Manuscript file type with the revised article:

Editorial Policy Policy requirements (Download the link to your computer as a PDF.)

- Behavioural and social science
- Ecological, evolutionary & environmental sciences
- Life sciences

<https://www.nature.com/documents/nr-reporting-summary.zip>

For your information, you can find some guidance regarding format requirements summarized on the following checklist:(<https://www.nature.com/documents/commsj-phys-style-formatting-checklist-article.pdf>) and formatting guide (<https://www.nature.com/documents/commsj-phys-style-formatting-guide-accept.pdf>).

REVIEWER COMMENTS:

Reviewer #1 (Remarks to the Author):

Wang and co-authors studied methane cycling microorganisms in intact and degraded permafrost soils using 16S amplicon sequencing. By combining data from multiple campaigns, they have developed a large pan-arctic dataset (10 sites, >600 individual samples). They combine this large dataset with samples from a permafrost degradation gradient in Alaska that covered wet and dry degraded endpoints. The authors find that permafrost settings host diverse communities of methanogens capable of multiple methanogenic pathways, whereas low affinity methanotrophs were largely limited to a single species. The effect of degradation on the methane cycle was highly affected by hydrological conditions, with wet conditions leading to an increase methanogen and low affinity methanotroph abundance while dry conditions lead to an increase in high affinity (atmospheric) methanotrophs.

My background is in biogeochemistry rather than microbial ecology, so I'm not fully capable of assessing the study with regards to its methodological details or its significance in the microbiological field. That being said, this is clearly a highly relevant study that addresses the microbiological side of an urgent problem (the impact of permafrost degradation on the future climate system), and to the degree that I can assess them the study's methods are at the state-of-the-art. The authors findings are important, in particular as they identify a single species dominating the pan-arctic low-affinity methanotrophy – this can be an important target for further physiological studies that will provide important inputs to arctic methane cycle models.

In contrast, I find the biogeochemical aspects of the manuscript weakly developed, at times wrong, and in general ignorant of a large body of literature that has already been developed long before molecular microbiology approaches. In principle, this would not be an important issue for a primarily microbiology focused paper, but the authors chose to frame the significance of their work in the context of the global biogeochemical implications of their results, so these should be appropriately handled.

Two statements about high affinity methanotrophs stand out as obviously wrong:

(1) In L321-327, the authors claim that atmospheric methane oxidation is an important contributor to SOC formation

through the provision of additional CO₂ to autotrophs. First, I would argue that in all known soils this is a quantitatively unimportant process (CO₂ production in soil is small compared to e.g. diffusion of CO₂ into the soil). If the authors want to claim this mechanistic relationship, some quantitative model of the impact of atmospheric methane oxidation on SOC formation needs to be added to prove that this mechanism is indeed possible (e.g., by how much does methane oxidation increase soil CO₂ concentrations, and how much does such an increase stimulate carbon fixation). To me, the positive correlation of relative abundance of methanotroph with SOC is a result of higher SOC being favourable to methanotroph growth rather than vice versa.

(2) In L328-330, the authors claim that increased atmospheric methanotroph abundance will mitigate atmospheric CH₄ concentrations. This is a common claim in studies of methane oxidation in upland soils, however it does not hold true from the perspective of global biogeochemical cycles. The atmospheric lifetime of methane is only ca. 10 years. Unlike CO₂, the atmospheric methane concentrations is therefore not driven by accumulation over time, but by the steady state equilibrium between production and consumption rates, which is reached on a decadal time scale. In these contexts, atmospheric concentrations stabilize at a level that is proportional to the global source strengths and inverse proportional to the global sink strengths (i.e., concentrations have risen over the last century not because of accumulation of CH₄ in the atmosphere, but because CH₄ emissions rates have continuously increased. From a climate perspective, a step change in emission rates for CH₄ is equivalent to a one-time emission of CO₂). Soil oxidation accounts for only the ca. 8% of the global methane sink (ca. 92% of methane are removed by atmospheric oxidation). If CH₄ oxidation in arctic soils increases the global soil CH₄ sink by 10%, this would only increase the CH₄ sink strength by 0.8% (and decrease CH₄ concentrations by the same amount) – something that can hardly be seen as mitigation given that atmospheric CH₄ concentrations have increased by >150% since pre-industrial times.

More generally, I think the article poorly incorporates much of the biogeochemical literature. Findings that water-logged conditions stimulate the abundance of methanogen and indirectly methanotrophs appear trivial to the reviewer, and speculations about how changes in the abundance of these functional groups affects the global CH₄ cycle upon permafrost thaw have been established long before molecular tools allowed their identification. More recently, e.g. Voigt et al (<https://www.nature.com/articles/s41558-023-01785-3>) proposed increased methane uptake of upland arctic soils. I find that if the authors want to express the significance of their findings in the context of how they will affect the arctic CH₄ cycle, they should (1) presenting what is already known about these dynamics based on previous work (using other methods than molecular microbiology) and (2) how their results improve our understanding beyond this existing knowledge. Like, we already knew how methane production and methane oxidation rates respond to environmental parameters, and we likely have better ways to study these responses than a microbial census (e.g. incubation assays). I believe that the detailed understanding of the identity of the microbial taxa that mediate these processes provided in this study is novel and useful, but I think the authors do not make it clear how this develops a process understanding beyond the status quo. Now, not every paper on environmental microbiology needs to provide such a better process understanding, but if that's not the goal I think the manuscripts aims should be stated differently.

After all this criticism, let me reiterate that I think this paper does contain a very important large-scale re-analysis, and it would be great to see a revised version published in your journal.

Reviewer #2 (Remarks to the Author):

General comments:

This is an interesting study on the distribution and abundance of methane-cycle associated microbiomes across the pan-Arctic. The authors effectively describe the diversity and ecological features of 16S-based phylotypes of methanogens and methanotrophs across broad spatial and temporal scales, including vertical horizons and extensive sample coverage. As stated in the manuscript, I agree that these microbial insights are crucial for future studies on methane dynamics in a warming Arctic. I have a major concern about the misuse of species-level identification based on short 16S sequences and suggest to avoid species-level assignment and interpretation across the manuscript. Below are some minor suggestions to improve the manuscript.

Specific comments:

The reliance on short-read 16S sequences may not be sufficient for species-level identification. In both abstract and results, authors keep mentioning the species names such as *M. lacus* and *M. tundripaludum*. Since genomic-level species resolution has not been established (even 100% identity in 100-200bp 16S fragment to a cultured species is not conclusive). I recommend avoiding species-level taxonomy throughout the text. Instead, using 'Methanosarcina-like' or 'Methylobacter-like' would be more appropriate.

There appears to be a distinction between 621 intact permafrost samples and Alaska site samples. Are permafrost-degraded sites across hydrological gradients only found in this Alaska site? Was the Alaska site specifically included because other pan-Arctic samples lacked hydrological gradient sites? Clarifying this would improve the context.

It seems like you re-used or downloaded 16S data which is already published in Varsadiya et al. (2021) rather than generating new 16S amplicon sequencing data. Please clearly distinguish between downloaded 16S datasets and those uniquely sequenced in this study.

Line 157 Does DN refer to DON or total dissolved nitrogen (TDN)?

Line 162 The 16S primer and region information for 100bp PE sequencing is missing. If the same primer set

515F/806R was used for 100bp PE, wouldn't 100 bp PE sequencing be too short for the 16S V4 region?

Line 168 Why didn't you use SILVA138.2, the latest version for taxonomic assignment? You used SILVA 138.1 for Alaska data shown in line 193. Please maintain consistency in the use of reference databases.

Line 186 Both ASVs and ZOTUs are similar but I'm not convinced why authors have used two different ASV-inferring approaches in one study. To make them comparable, using a single analytical pipeline to infer ASVs would be more intuitive?

Line 199 It appears that 200-300bp 16S sequences were used for phylogenetic tree construction. Isn't this too short for a robust phylogenetic analysis?

Line 205 Please specify key parameters such as the evolutionary model used in IQ-TREE?

Line 230, 232, 239, 240 The results are presented in Fig. 3, not Fig. 1.

Line 233 According to the Fig. S1, the positive correlation between water content and methanogen relative abundance is only true for the organic layer dataset, not for other categories. Therefore, you should not generalize the finding applicable to entire soil horizons.

Line 243 It is intuitively true but Fig. S1e shows two diverging patterns: while MT abundance is high in subsoil and cryoOM (where low and moderate level of MG abundance is found), MT abundance remains low or moderate levels in organic layer and permafrost, where MT abundance is high. Given that correlation coefficients are below 0.3, this trend is not strong enough to make definitive conclusions.

Line 267 The claim that 6 phylotypes belong to *M. tundripaludum* based on 200-300bp 16S sequences is not robust. Instead, refer to them as Methylobacter-like sequences. For reference, this isolated methylobacter sp. strain clustered separately from *M. tundripaludum* although their 16S identity is 99% (<https://academic.oup.com/ismecommun/article/2/1/85/7460599>).

Line 278-280 The low methanogen abundance in the Intact site (2019 and 2021) may be due to shallower sampling depths compared to Dry-Intermediate-Wet sites.

Line 281-282 The four phylotypes do not necessarily belong to *M. luminyensis* and may represent different genera or families. How are you sure they are associated with *M. luminyensis*?

Line 310-317 The Dry site (2019) contained some methanogens in deeper layers, but anaerobic and type I MOB were absent. Also, low abundance was detected in 46-60cm depth in the Wet site in 2021 but conventional MOB was detected in deeper layers of 91-105cm depth. Therefore, these results do not strongly support the claim that conventional methanotroph abundance depends on methanogen abundance at each site.

Line 321-324 These patterns apply only to 2019 samples, not 2021 samples. Therefore, you should clearly specify here.

Line 326 This statement, CO₂ converted by type II MOB is directly consumed by autotrophic carbon fixers, seems speculative. Without quantitative evidence, this claim should be avoided. What if SOC-derived organic carbon substrates may support Type II MOB growth and activity?

Line 332-350 Again, these phylotypes could be neighbors of *M. tundripaludum* SV9. However, they should be assumed to be the same species.

Line 338-341 The phylogenetic placement of these sequences is uncertain based on short-read data. The 16S sequences from *M. tundripaludum* MAGs were selectively included for the tree. How about adding more 16S sequences from Methylobacter-like MAGs in GTDB database or MAGs used in <https://academic.oup.com/ismecommun/article/2/1/85/7460599> for tree construction? If these phylotypes are uniquely formed a single cluster with 16S sequences obtained from *M. tundripaludum* MAGs, even if these additional reference sequences were included, their placement would be more convincing.

Line 352-357 100% identity between ASVs and ZOTUs based on 200-300bp 16S sequences does not mean that all of them belong to *M. tundripaludum*. I strongly recommend using Methylobacter-like phylotypes instead of *M. tundripaludum*. Therefore, I cannot agree with *M. tundripaludum* as a single species dominated in pan-Arctic permafrost-affected soils.

Line 364 In Fig 5a, only four more phylotypes were detected in Dry-Intermediate-Wet sites compared to the Intact site, all belonging to Methanomassiliococcales. Given this small number of additional phylotypes, it is not justified to state 'methanogen richness increased due to permafrost degradation'.

Line 386-388 The claim that *M. tundripaludum* is a single dominant species across pan-Arctic permafrost-affected soils is not supported by the data.

Figures

Fig. 6 Add spacing between Type and I/II, such as Type I and Type II.

Fig. 7 and Fig. S5 The outgroup taxa used in both trees are too evolutionarily distant from ingroup taxa. Please use appropriate outgroup taxa and regenerate the trees.

Reviewer #3 (Remarks to the Author):

This manuscript describes a survey of the distribution, relative abundance, and diversity of methanogen and methanotrophic populations in different arctic regions. The results and interpretation of the data are solid and presents some novel results in showing the dominance of certain species of methanotrophs, effect of soil moisture content as a key driver of methanotrophic and methanogenic composition, and, hints at the potential role of atm methane oxidizers, as key players in the future methane/CO₂ cycles in future drying permafrost scenarios. The manuscript is solid mostly well written with clear objectives and sound statistical approaches. Nevertheless, there are some recommendations of revisions as indicated below.

Major recommendations

1. A strength of the manuscript is the number (621) of samples tested from 9 sites across the Arctic regions. The chemical /physical analyses performed and the statistical linkage to the 16S genes from these sites is appropriate. However, there is no in situ CH₄/CO₂ flux data given to support the overall the interpretations of the authors. For example, one would have expected the “dry” permafrost samples containing the atmospheric methane oxidizers to have become methane and CO₂ sinks?

2. While the authors have mostly focused on their 10 samples and 16S sequencing, there has been multiple reports over the last ~15 years on other sites where 16S and metagenomic sequence data sets are available for comparison such as the numerous reports on this topic from Axel Heiberg Island and Ellesmere Island in the Canadian high arctic. See for example, Altshuler et al. 2022. Scientific Reports (<https://doi.org/10.1038/s41598-021-04486-z>). Indeed, the novelty of the findings of this manuscript could be better highlighted. The interpretations / conclusions presented are mostly focused on the sites described in this study but could also be interpreted in already published papers on this topic.

3. I understand why a 16S rRNA approach was only used in this study but was puzzled by why the study did not also look for specific gene markers for methanotrophs (ex. mmoX, pmoA) and methanogens ex. mcrA) which would have supported the 16S results, especially since the 16S sequencing analyses only used a short 250 bp fragments? Similarly, metagenomic sequencing of at least some of the samples would have also enhanced, this study, especially, the comparison of the Alaska wet, intermediate, and dry sites?

Minor recommendations

1. If possible please indicate the time of sampling for each of the sampling sites as populations of these microbiomes may have shifted between the winter season and summer season?

2. Some of the terminology used could be more concise please

L. 252 “prevalently”?

L. 274 “adjustments” could be changed to adaptations and, if possible, the major adaptations that were found?

L. 344 “certain’?

L. 113-114 “permafrost sites around the north pole”? should be changed to Arctic permafrost sites as the “north pole” is not accurate.

4. Figure 1 Panel A is difficult to read and interpret with the different color schemes and the poor quality of the map showing the different sampling sites. What, for example, does the white color indicate? Please also move the Panel Labels (ie A, B, C, etc) to the beginning of Panel description from the end (Please do this for each figure with panels as it makes it easier for the reader?

Communications Earth & Environment is committed to improving transparency in authorship. As part of our efforts in this direction, we are now requesting that all authors identified as ‘corresponding author’ create and link their Open Researcher and Contributor Identifier (ORCID) with their account on the Manuscript Tracking System prior to acceptance. ORCID helps the scientific community achieve unambiguous attribution of all scholarly contributions. You can create and link your ORCID from the home page of the Manuscript Tracking System by clicking on ‘Modify my Springer Nature account’ and following the instructions in the link below. Please also inform all co-authors that they can add their ORCIDs to their accounts and that they must do so prior to acceptance.
<https://www.springernature.com/gp/researchers/orcid/orcid-for-nature-research>

Version 1:

Decision Letter:

Dear Dr Wang,

Your manuscript titled "Methane-cycling microbiomes in soils of the pan-Arctic and their response to permafrost degradation" has now been seen by our reviewers, whose comments appear below. In light of their advice we are delighted to say that we are happy, in principle, to publish a suitably revised version in Communications Earth & Environment.

We therefore invite you to revise your paper one last time to address the remaining concerns of our reviewers. At the same time we ask that you edit your manuscript to comply with our format requirements and to maximise the accessibility and therefore the impact of your work.

EDITORIAL REQUESTS:

******Please take care to match our formatting and policy requirements. We will check revised manuscript and return manuscripts that do not comply. Such requests will lead to delays. ******

SUBMISSION INFORMATION:

OPEN ACCESS:

Communications Earth & Environment is a fully open access journal. Articles are made freely accessible on publication. For further information about article processing charges, open access funding, and advice and support from Nature Portfolio, please visit <https://www.nature.com/commsenv/open-access>

Link Redacted

Best regards,

Somaparna Ghosh, PhD
Associate Editor,
Communications Earth & Environment
Consulting Editor,
Communications Sustainability

REVIEWERS' COMMENTS:

Reviewer #1 (Remarks to the Author):

This is the revised version of a manuscript I reviewed previously. Overall, the authors have done a great job at addressing all reviewer comments and I think the manuscript can be published in the current form.

I have some minor points:

Fig S1: I think the axis in this figure need to be switched - you want to predict MT/MG RA based on pH etc., not the other way around (so the uncertainty needs to be in terms of MT/MG). This should not affect R values or significance.

L401-402: "our data suggest that atmMOB might play a pivotal role in increasing the CH₄ sink capacity in drained landscapes after permafrost thaw in a future warmer climate." This sounds like a bit of a tautology to me - isn't CH₄ (net soil CH₄) sink by definition characterized by atmMOB? maybe reword this.

Reviewer #2 (Remarks to the Author):

The authors have sufficiently addressed the issues and concerns (e.g., 16S-based taxonomic assignment and various methodological issues, etc.) I raised in the first-round review. I have no further comments on the revised manuscript.

Reviewer #3 (Remarks to the Author):

The authors have addressed many of the concerns of this paper appropriately. Some of their responses are somewhat weak, especially the comments on why metagenomic approaches were not used. Nevertheless, I think this paper contains sufficient significant results and novelty.

** Visit Nature Portfolio's author and referees' website at <http://www.nature.com/authors> for information about policies, services and author benefits**

Response to Reviewers' Comments

Thank you for taking the time to write thoughtful responses to our manuscript. We appreciate your effort, and your suggestions have greatly helped in strengthening our paper. We have worked to incorporate your suggestions and provided detailed responses to each of your comments below. In addition, we include three general responses to comments that were addressed by multiple reviewers.

General comments:

(1) Correlations in Fig. S1 are not strong and the statistics should be calculated based on each layer (Reviewer 1 and 2).

GENERAL ANSWER 1: We re-calculated the correlation statistics based on each layer. We found that water content was significantly and positively correlated in organic layer, cryoOM and permafrost (Fig. S1a) in which the methanogens were also abundant (Fig. 1c). Due to the changes of the statistics, we re-wrote this part (related to **Comment L233** of Reviewer 2) as following:

“The methanogen distribution along the vertical profiles was often location-specific, sometimes increasing with depth, and sometimes having the highest abundance in the organic layers (Fig.3c and S2). In mineral top- and subsoils, methanogens were lowly abundant likely due to the lack of substrates. Nevertheless, their relative abundance and soil water content were significantly positively correlated in organic layer, cryoOM and permafrost (Fig. S1a) in which the methanogens were also abundant (Fig. 3c). This indicates that the hydrologic status is one key driving factor for methanogen abundance by limiting oxygen diffusion into the soil profile in layers where they habitat. pH only showed a significant and negative correlation with methanogen relative abundances in the permafrost layer (Fig. S1c).” (L228-236)

Further, we found improved correlations between methanogen and methanotroph abundances in all layers except the permafrost layer (Fig. S1e). Accordingly, we re-wrote this part (related to **Comment L243** of Reviewer 2) as following:

“Along the soil profiles, methanotrophs were less abundant in the organic layer and topsoil compared to the other deeper layers (Fig. 3d), but this pattern again varied between sites (Fig. S3). Compared to methanogens, water content only showed a weak but still significant correlation with methanotroph abundance in the organic layer (Fig. S1b). Like methanogens, methanotroph

abundance was negatively and significantly correlated with pH only in the permafrost layer (Fig. S1d). Overall, methanotroph abundances showed a strong and positive correlation with methanogen abundances in all layers except permafrost (Fig. S1e). This suggests that these methanotrophs in the active layers likely rely on the CH₄ produced by the methanogens. While methanogen abundances could be mainly driven by abiotic factors such as water (Fig. S1a), methanotroph abundances could be driven by the resource availability in pan-Arctic permafrost affected soils.” (L240-250)

(2) Incorporation of literatures with biogeochemical perspectives or more microbiome perspectives in degraded sites (Reviewer 1 and 3)

GENERAL ANSWER 2: We went through many literatures including Voigt et al. and Altshuler et al. 2022 regarding CH₄ dynamics in wet and dry Arctic landscapes. Based on that, we added one paragraph in the Introduction addressing the current knowledge about CH₄ emission and consumption in Arctic and pointed out the knowledge gap that the microbiomes driving these patterns are under-characterized.

“Arctic wetlands and water-logged soils are known hotspots of CH₄ emissions^{42,43}. This is due to the wet conditions after permafrost thaw that contribute to the increase of methanogen abundances and the shift in their community compositions^{44,45}. A substantial number of studies have particularly addressed the consumption of atmospheric CH₄ in well-drained Arctic soils^{12,46-50}, but surprisingly little attention has been paid to the microbiomes driving this phenomenon. Only a few studies mentioned atmMOB as the main drivers of CH₄ consumption in Arctic soils^{37,51,52}. Moreover, studies comparing the abundance and composition of methane-cycling microbiomes across soil moisture gradients following permafrost thaw remain scarce.” (L98-105)

Further, we added a new section “**Factors driving methane-cycling microbiome changes in intact and degraded sites**” (L339-382) in the discussion. Here, we discussed the impact of edaphic factors (moisture, temperature, pH and SOC) on the microbiomes based on the correlations. We compared our study with previous studies finding the same correlations but with CH₄ fluxes and indicated that our study provided the microbiome evidence for these process data based observations. We also provided some, speculative, explanations of these correlations. We finally suggested that both abiotic and biotic factors (methanogen and methanotroph abundances and

compositions) should be considered in future modelling for predicting Arctic CH₄ dynamics.

Regarding more microbiome perspectives in degraded sites, we screened for publications and found that few studies monitored microbiomes therein, despite that many studies found that dry landscapes are hot spots for CH₄ sink in these regions. Our findings regarding atmMOB could support these studies. This is also another novelty of our study.

(3) Clarification on why degraded sites were only included in Alaksa study. (Reviewer 2 and 3)

GENERAL ANSWER 3: In this pan-Arctic project, we mainly focused on the description of microbiomes across the pan-Arctic, while we mainly investigated the impact of different water status after permafrost thaw on microbiomes in the Alaska project. Therefore, we did not look for the hydrological gradient of sites in other Arctic regions. Nonetheless, we think that adding the perspective of permafrost degradation and its impact on methane-cycling microbiome could help us better understand future possible changes for methane dynamics. Indeed, a case study in Alaska would not represent a pan-Arctic view. Therefore, we changed the title to “Methane-cycling microbiomes in soils of the pan-Arctic and their response to permafrost degradation” to avoid the confusion. We also mentioned in the new version that more investigations from other Arctic regions are needed to support our results. For instance,

“More investigations of methanotroph communities in the well drained soils from other Arctic regions are needed to further understand their CH₄ sink capacity, particularly in times of climate change when more dry conditions are expected.” (L380-382)

“Although investigations from other Arctic regions are needed for validation, our data suggest that atmMOB might play a pivotal role in increasing the CH₄ sink capacity in drained landscapes after permafrost thaw in a future warmer climate.” (L400-402)

Reviewer #1 (Remarks to the Author):

Wang and co-authors studied methane cycling microorganisms in intact and degraded permafrost soils using 16S amplicon sequencing. By combining data from multiple campaigns, they have developed a large pan-arctic dataset (10 sites, >600 individual samples). They combine this large dataset with samples from a permafrost degradation gradient in Alaska that covered wet and dry

degraded endpoints. The authors find that permafrost settings host diverse communities of methanogens capable of multiple methanogenic pathways, whereas low affinity methanotrophs were largely limited to a single species. The effect of degradation on the methane cycle was highly affected by hydrological conditions, with wet conditions leading to an increase methanogen and low affinity methanotroph abundance while dry conditions lead to an increase in high affinity (atmospheric) methanotrophs.

My background is in biogeochemistry rather than microbial ecology, so I'm not fully capable of assessing the study with regards to its methodological details or its significance in the microbiological field. That being said, this is clearly a highly relevant study that addresses the microbiological side of an urgent problem (the impact of permafrost degradation on the future climate system), and to the degree that I can assess them the study's methods are at the state-of-the-art. The authors findings are important, in particular as they identify a single species dominating the pan-arctic low-affinity methanotrophy – this can be an important target for further physiological studies that will provide important inputs to arctic methane cycle models.

In contrast, I find the biogeochemical aspects of the manuscript weakly developed, at times wrong, and in general ignorant of a large body of literature that has already been developed long before molecular microbiology approaches. In principle, this would not be an important issue for a primarily microbiology focused paper, but the authors chose to frame the significance of their work in the context of the global biogeochemical implications of their results, so these should be appropriately handled.

Two statements about high affinity methanotrophs stand out as obviously wrong:

(1) In L321-327, the authors claim that atmospheric methane oxidation is an important contributor to SOC formation through the provision of additional CO₂ to autotrophs. First, I would argue that in all known soils this is a quantitatively unimportant process (CO₂ production in soil is small compared to e.g. diffusion of CO₂ into the soil). If the authors want to claim this mechanistic relationship, some quantitative model of the impact of atmospheric methane oxidation on SOC formation needs to be added to proof that this mechanisms is indeed possible (e.g., by how much

does methane oxidation increase soil CO₂ concentrations, and how much does such an increase stimulate carbon fixation). To me, the positive correlation of relative abundance of methanotroph with SOC is a result of higher SOC being favourable to methanotroph growth rather than vice versa.

ANSWER: Thanks for your critique. We agree that the positive correlation rather indicates that higher SOC favors methanotroph growth. We revised this discussion accordingly and further added some possible explanations. The new discussion is as following:

“The correlations further suggested that higher SOC content may favor the proportion of type II MOB (Fig. 6). This is because higher SOC input can provide alternative carbon substrates that enhance their growth as many type II MOB are facultative mixotrophs that can grow on low-molecular-weight organic compounds³⁰. This is in line with a recent study that found an identical relationship of type II MOB with SOC in paddy soils⁷⁰.” (L362-366)

(2) In L328-330, the authors claim that increased atmospheric methanotroph abundance will mitigate atmospheric CH₄ concentrations. This is a common claim in studies of methane oxidation in upland soils, however it does not hold true from the perspective of global biogeochemical cycles. The atmospheric lifetime of methane is only ca. 10 years. Unlike CO₂, the atmospheric methane concentrations is therefore not driven by accumulation over time, but by the steady state equilibrium between production and consumption rates, which is reached on a decadal time scale. In these context, atmospheric concentrations stabilize at a level that is proportional to the global source strengths and inverse proportional to the global sink strengths (i.e., concentrations have risen over the last century not because of accumulation of CH₄ in the atmosphere, but because CH₄ emissions rates have continuously increased. From a climate perspective, a step change in emission rates for CH₄ is equivalent to a one-time emission of CO₂). Soil oxidation accounts for only the ca. 8% of the global methane sink (ca. 92% of methane are removed by atmospheric oxidation). If CH₄ oxidation in arctic soils increases the global soil CH₄ sink by 10%, this would only increase the CH₄ sink strength by 0.8% (and decrease CH₄ concentrations by the same amount) – something that can hardly be seen as mitigation given that atmospheric CH₄ concentrations have increased by >150% since pre-industrial times.

ANSWER: Thanks for your critique. We deleted this interpretation according to the changes we

made regarding your **Comment (1)**. Moreover, we revised some related statements, e.g., from “mitigation of CH₄” to “increase of CH₄ sink capacity” (e.g., L401), which we think is more precise based on microbiome abundance data.

More generally, I think the article poorly incorporates much of the biogeochemical literature. Findings that water-logged conditions stimulate the abundance of methanogen and indirectly methanotrophs appear trivial to the reviewer; and speculations about how changes in the abundance of these functional groups affects the global CH₄ cycle upon permafrost thaw have been established long before molecular tools allowed their identification. More recently, e.g. Voigt et al (<https://www.nature.com/articles/s41558-023-01785-3>) proposed increased methane uptake of upland arctic soils. I find that if the authors want to express the significance of their findings in the context of how they will affect the arctic CH₄ cycle, they should (1) presenting what is already known about these dynamics based on previous work (using other methods than molecular microbiology)

ANSWER: Please refer to **GENERAL ANSWER 1**. We explained better on the correlations showing positive relationship between water content and methanogens, and between methanogens and methanotrophs. Please refer to **GENERAL ANSWER 2** for what we have added by incorporating more literatures.

and (2) how their results improve our understanding beyond this existing knowledge. Like, we already knew how methane production and methane oxidation rates respond to environmental parameters, and we likely have better ways to study these responses than a microbial census (e.g. incubation assays). I believe that the detailed understanding of the identity the microbial taxa that mediate these processes provided in this study is novel and useful, but I think the authors do not make it clear how this develops a process understanding beyond the status quo. Now, not every paper on environmental microbiology needs to provide such a better process understanding, but if that's not the goal I think the manuscripts aims should be stated differently.

ANSWER: We have incorporated these important comments. Please refer to the **GENERAL ANSWER 2**. Moreover, since our study provides no actual process data and our abundance is relative, we should not over-interpret our data beyond our main focus which is the taxonomy

composition. However, we did provide some discussion on how our data can provide a process understanding on CH₄ dynamics in a future climate in the last session of our discussion. Please refer to “**The methane-cycling microbiome in Arctic permafrost: unlocking future climate scenarios**” (L384-402)

After all this criticism, let me reiterate that I think this paper does contain a very important large-scale re-analysis, and it would be great to see a revised version published in your journal.

Reviewer #2 (Remarks to the Author):

General comments:

This is an interesting study on the distribution and abundance of methane-cycle associated microbiomes across the pan-Arctic. The authors effectively describe the diversity and ecological features of 16S-based phylotypes of methanogens and methanotrophs across broad spatial and temporal scales, including vertical horizons and extensive sample coverage. As stated in the manuscript, I agree that these microbial insights are crucial for future studies on methane dynamics in a warming Arctic. I have a major concern about the misuse of species-level identification based on short 16S sequences and suggest to avoid species-level assignment and interpretation across the manuscript. Below are some minor suggestions to improve the manuscript.

Specific comments:

*The reliance on short-read 16S sequences may not be sufficient for species-level identification. In both abstract and results, authors keep mentioning the species names such as *M. lacus* and *M. tundripaludum*. Since genomic-level species resolution has not been established (even 100% identity in 100-200bp 16S fragment to a cultured species is not conclusive). I recommend avoiding species-level taxonomy throughout the text. Instead, using ‘*Methanosarcina*-like’ or ‘*Methylobacter*-like’ would be more appropriate.*

ANSWER: Thanks for commenting on this important point. We agree with you. We removed the discussion about the species level identity and avoided mentioning the species name through the manuscript and in the figures. The information of the NCBI BLASTN based identification is still

provided in Table S2 to verify the genus level taxonomy assignment derived from the latest SILVA database. We only kept one discussion regarding *Methylobacter tundripaludum* but with serious cautions, as following:

“Further, NCBI BLASTN results suggested that these *Methylobacter*-like phylotypes were all closely related to *M. tundripaludum* SV96 (95%-100% identity, Table S2) which was first isolated from Svalbard peatlands⁶⁴. This hints that the pan-Arctic microbial CH₄ filter might even be dominated by one single species, although the species-level identity is not robust based a short fragment (250 bp) of 16S rRNA gene. *M. tundripaludum* was also found as dominant methanotrophs in other locations on Svalbard⁶⁵ and in the Lena Delta in Siberia⁶⁶, which supports our finding. Moreover, *M. tundripaludum* was found as an active methane oxidizer in high Arctic wetlands using stable isotope probing⁶⁷. A recent study showed physiological adaptations of *M. tundripaludum* to varying temperatures by adjusting their central metabolism, protein biosynthesis, cell walls and storage⁶⁸. This might explain the prevalence and dominance of *M. tundripaludum* across heterogeneous Arctic regions.” (L275-286)

There appears to be a distinction between 621 intact permafrost samples and Alaska site samples. Are permafrost-degraded sites across hydrological gradients only found in this Alaska site? Was the Alaska site specifically included because other pan-Arctic samples lacked hydrological gradient sites? Clarifying this would improve the context.

ANSWER: Only the Alaska sites were specifically chosen for the degraded permafrost conditions. All other sites were sampled in earlier sampling campaigns, in a context where no permafrost degradation was specifically sampled. At that time, the cryoturbation and depth-resolved sampling was in the focus of the scientific projects. Please refer to **GENERAL ANSWER 3**. We hope that this answer and the edits done in the manuscript do clarify the context.

It seems like you re-used or downloaded 16S data which is already published in Varsadiya et al. (2021) rather than generating new 16S amplicon sequencing data. Please clearly distinguish between downloaded 16S datasets and those uniquely sequenced in this study.

ANSWER: Thanks for your suggestions. We have added two sentences to clarify this in the methods session. In summary, the data from the intact pan-Arctic permafrost sites were previously

published, while the Alaska data are from our study.

“The sequencing data of the pan-Arctic samples were generated and published by previous studies⁵³⁻⁵⁶.” (L170-171)

“The sequencing data of Alaska samples was generated for this study and has not been published elsewhere.” (L191-192)

Line 157 Does DN refer to DON or total dissolved nitrogen (TDN)?

ANSWER: Thank you for spotting this inconsistency. DN refers to dissolved total nitrogen (DTN), as we did not measure and subtract inorganic N-forms like NH_4^+ or NO_3^- . We changed the terminology accordingly.

Line 162 The 16S primer and region information for 100bp PE sequencing is missing. If the same primer set 515F/806R was used for 100bp PE, wouldn't 100 bp PE sequencing be too short for the 16S V4 region?

ANSWER: Yes, 100 bp PE is too short for the V4 region. This is why we could not merge the forward and reverse sequences for 100 bp PE sequencing data. Instead, we only used the forward sequences and trimmed them all to 100 bp for comparability. We revised this sentence to “To make all sites comparable, forward sequences were all trimmed to 100 bp and then used for downstream analysis.” (L171-172)

Line 168 Why didn't you use SILVA138.2, the latest version for taxonomic assignment? You used SILVA 138.1 for Alaska data shown in line 193. Please maintain consistency in the use of reference databases.

ANSWER: We re-assigned the taxonomy of both ZOTUs and ASVs against SILVA138.2 using a Naive Bayesian Classifier algorithm implemented in *dada2* pipeline. Now the taxonomies of both phylotypes are consistent. We further identified the taxonomy assignment with NCBI BLASTN (Table S2). The changes can be found in L174-181.

Line 186 Both ASVs and ZOTUs are similar but I'm not convinced why authors have used two different ASV-inferring approaches in one study. To make them comparable, using a single

analytical pipeline to infer ASVs would be more intuitive?

ANSWER: The pan-Arctic data were analyzed before based on 100 bp for comparability with our previous studies, while the Alaska data were newly acquired and were analyzed based on 250 bp. Both USEARCH and DADA2 provide high resolution on phylotypes and should provide identical results for at least genus identification. This is fine with our data since we don't interpret our data based on species level any more in the new version. Also, when we made the comparison, we always compared their taxonomy (not the number of phylotypes), which is not sensitive to the phylotype-generating methods (at least not at the genus level). Moreover, the local BLASTN also indicates that most of the ASVs and ZOTUs have 100% identity, and their taxonomy assignments are also matching (Table S3). To avoid the confusion, we now used the term 'phylotypes' in the Results and Discussion instead of ZOTUs or ASVs in the new manuscript. We hope that we have convinced you that it is not necessary to re-run the whole analysis for comparability.

Line 199 It appears that 200-300bp 16S sequences were used for phylogenetic tree construction. Isn't this too short for a robust phylogenetic analysis?

ANSWER: We agree that 200-300 bp 16S rRNA gene might be too short for phylogenetic tree construction. We have therefore removed the whole part related with phylogenetic tree reconstruction because this tree was originally built for identifying the species within *Methylobacter*. As you mentioned above that 200-300 bp is too short for species identification, there is no need any more to keep that tree.

Line 205 Please specify key parameters such as the evolutionary model used in IQ-TREE?

ANSWER: We removed the tree based on your **Comment Line 199**.

Line 230, 232, 239, 240 The results are presented in Fig. 3, not Fig. 1.

ANSWER: We made the changes. Thank you!

Line 233 According to the Fig. S1, the positive correlation between water content and methanogen relative abundance is only true for the organic layer dataset, not for other categories. Therefore, you should not generalize the finding applicable to entire soil horizons.

ANSWER: Please refer to **GENERAL ANSWER 1.**

Line 243 It is intuitively true but Fig. S1e shows two diverging patterns: while MT abundance is high in subsoil and cryoOM (where low and moderate level of MG abundance is found), MT abundance remains low or moderate levels in organic layer and permafrost, where MT abundance is high. Given that correlation coefficients are below 0.3, this trend is not strong enough to make definitive conclusions.

ANSWER: Please refer to **GENERAL ANSWER 1.**

*Line 267 The claim that 6 phylotypes belong to *M. tundripaludum* based on 200-300bp 16S sequences is not robust. Instead, refer to them as *Methylobacter*-like sequences. For reference, this isolated *methylobacter* sp. strain clustered separately from *M. tundripaludum* although their 16S identity is 99% (<https://academic.oup.com/ismecommun/article/2/1/85/7460599>).*

ANSWER: Thank you for this important comment. **As replied to your first Specific Comment**, we now refer to them as *Methylobacter*-like phylotypes throughout the manuscript. We avoided highlighting the species identity in the new version.

Line 278-280 The low methanogen abundance in the Intact site (2019 and 2021) may be due to shallower sampling depths compared to Dry-Intermediate-Wet sites.

ANSWER: Thanks for the hint. We added some discussion regarding the sampling depth, as following:

“The low abundance in the Intact site might be due to the shallower sampling depth as more methanogens were found in the layers lower than 40-60 cm in the other sites (Fig. 5a). The deeper layers in the Intact site were frozen permafrost in which microbes should not be active despite their existence.” (L295-298)

*Line 281-282 The four phylotypes do not necessarily belong to *M. luminyensis* and may represent different genera or families. How are you sure they are associated with *M. luminyensis*?*

ANSWER: We removed the species level identity of Methanomassiliicoccales as well. We now refer them to “4 phylotypes of the order Methanomassiliicoccales” (L292).

Line 310-317 The Dry site (2019) contained some methanogens in deeper layers, but anaerobic and type I MOB were absent. Also, low abundance was detected in 46-60cm depth in the Wet site in 2021 but conventional MOB was detected in deeper layers of 91-105cm depth. Therefore, these results do not strongly support the claim that conventional methanotroph abundance depends on methanogen abundance at each site.

ANSWER: Thanks for your critique. We deleted the claim that conventional methanotroph abundance depends on methanogen abundance at each site. Instead, we argue that these MOB found in deep layers were either inactive or supported by other CH₄ sources (e.g., plant-associated or abiotic sources) in these sites. The new version reads like following:

“In contrast, the Intact and Wet sites only hosted “conventional” type I MOB in deeper layers although almost no methanogens were detected. In the Intermediate-Wet site, the dominant MOB switched from *Methylocapsa* in top layers to *Ca. Methyloirabilis* in middle layers then to *Methylobacter* in deep layers (Fig. 5b). The anaerobic and type I MOB found in the middle and deeper layers might be supported by the methanogens living in these layers (Fig. 5a). However, this relationship could not be identified in the other sites, suggesting either that these MOB were inactive or that other CH₄ sources (e.g., plant-associated or abiotic sources) supported them in these sites.” (L330-337)

Line 321-324 These patterns apply only to 2019 samples, not 2021 samples. Therefore, you should clearly specify here.

ANSWER: We revised this paragraph according to your comment, as following:

“Methanogens and type I MOB showed congruent responses to the soil physiochemical properties (Fig. 6). They both showed a positive correlation with depth and pH, but a negative correlation with SOC, TN, C/N and temperature in 2019 (Fig. 6a). In contrary, type II MOB positively correlated with SOC, TN, C/N and temperature, but negatively with depth, moisture and pH (Fig. 6a). In 2021, most correlations were insignificant (Fig. 6b). However, depth, moisture, pH, SOC, TN and C/N showed a same relationship with methanogens, type I and type II MOB, respectively, as compared to 2019 (Fig. 6a and 6b).” (L340-346)

Line 326 This statement, CO₂ converted by type II MOB is directly consumed by autotrophic carbon fixers, seems speculative. Without quantitative evidence, this claim should be avoided. What if SOC-derived organic carbon substrates may support Type II MOB growth and activity?

ANSWER: Thanks for pointing this out. This is in line with Comment (1) from Reviewer 1. We revised this discussion as following:

“The correlations further suggested that higher SOC content may favor the proportion of type II MOB (Fig. 6). This is because higher SOC input can provide alternative carbon substrates that enhance their growth as many type II MOB are facultative mixotrophs that can grow on low-molecular-weight organic compounds³⁰. This is in line with a recent study that found an identical relationship of type II MOB with SOC in paddy soils⁷⁰.” (L362-366)

Line 332-350 Again, these phylotypes could be neighbors of M. tundripaludum SV9. However, they should be assumed to be the same species.

ANSWER: We removed this part in the new version according to your previous comment that we should not implement species identification based on a short fragment of 16S rRNA gene.

Line 338-341 The phylogenetic placement of these sequences is uncertain based on short-read data. The 16S sequences from M. tundripaludum MAGs were selectively included for the tree. How about adding more 16S sequences from Methylobacter-like MAGs in GTDB database or MAGs used in <https://academic.oup.com/ismecommun/article/2/1/85/7460599> for tree construction? If these phylotypes are uniquely formed a single cluster with 16S sequences obtained from M. tundripaludum MAGs, even if these additional reference sequences were included, their placement would be more convincing.

ANSWER: According to your **first Specific Comment** and **Comment L199**, we removed the whole part related with phylogenetic tree because this tree was originally built for identifying the species within *Methylobacter*.

Line 352-357 100% identity between ASVs and ZOTUs based on 200-300bp 16S sequences does not mean that all of them belong to M. tundripaludum. I strongly recommend using Methylobacter-like phylotypes instead of M. tundripaludum. Therefore, I cannot agree with M. tundripaludum as

a single species dominated in pan-Arctic permafrost-affected soils.

ANSWER: As answered before, we avoided species level identity and used *Methylobacter*-like phylotypes as you suggested.

Line 364 In Fig 5a, only four more phylotypes were detected in Dry-Intermediate-Wet sites compared to the Intact site, all belonging to Methanomassillioccales. Given this small number of additional phylotypes, it is not justified to state ‘methanogen richness increased due to permafrost degradation’.

ANSWER: We agree with you. We revised this sentence as following:

“However, due to their low diversity and low proportions in microbiome in all sites (Fig. 5a and Fig. S4a), it was unclear whether permafrost thaw resulted in effective changes in methanogen communities.” (L388-390)

Line 386-388 The claim that M. tundripaludum is a single dominant species across pan-Arctic permafrost-affected soils is not supported by the data.

ANSWER: We now only refer them to *Methylobacter*-like/related phylotypes.

Figures

Fig. 6 Add spacing between Type and I/II, such as Type I and Type II.

ANSWER: Changed.

Fig. 7 and Fig. S5 The outgroup taxa used in both trees are too evolutionarily distant from ingroup taxa. Please use appropriate outgroup taxa and regenerate the trees.

ANSWER: According to previous comments, the phylogenetic tree part was removed.

Reviewer #3 (Remarks to the Author):

This manuscript describes a survey of the distribution, relative abundance, and diversity of methanogen and methanotrophic populations in different arctic regions. The results and interpretation of the data are solid and presents some novel results in showing the dominance of

certain species of methanotrophs, effect of soil moisture content as a key driver of methanotrophic and methanogenic composition, and, hints at the potential role of atm methane oxidizers, as key players in the future methane/CO2 cycles in future drying permafrost scenarios. The manuscript is solid mostly well written with clear objectives and sound statistical approaches. Nevertheless, there are some recommendations of revisions as indicated below.

Major recommendations

1. A strength of the manuscript is the number (621) of samples tested from 9 sites across the Arctic regions. The chemical /physical analyses performed and the statistical linkage to the 16S genes from these sites is appropriate. However, there is no in situ CH4/CO2 flux data given to support the overall the interpretations of the authors. For example, one would have expected the “dry” permafrost samples containing the atmospheric methane oxidizers to have become methane and CO2 sinks?

ANSWER: Our study focuses on methane-cycling microbiomes that are responsible for the production and consumption of CH₄. The core of this microbiology-focused study is the distribution of methanogens and methanotrophs at a pan-Arctic scale, and their response to permafrost thaw using Alaska sites as a case study. We agree that flux data could support our interpretations, but the key message of our microbiology-focused manuscript would still be valid without flux data support. Moreover, we always interpreted our data with cautions. We now refer to “CH₄ sink capacity” or “CH₄ dynamics” not directly to CH₄ fluxes in the interpretations.

While we have no CH₄ flux data in our study, we do provide some introductions and discussions based on the published literatures in which CH₄ fluxes were monitored in these landscapes. Please refer to **GENERAL ANSWER 2**. Please also refer to **GENERAL ANSWER 3**.

CO₂ fluxes are of little relevance to our study since methane-cycling microbiomes contribute little to CO₂ dynamics due to their low abundances.

2. While the authors have mostly focused on their 10 samples and 16S sequencing, there has been multiple reports over the last ~15 years on other sites where 16S and metagenomic sequence data sets are available for comparison such as the numerous reports on this topic from Axel Heiberg

Island and Ellesmere Island in the Canadian high arctic. See for example, Altshuler et al. 2022. Scientific Reports (<https://doi.org/10.1038/s41598-021-04486-z>). Indeed, the novelty of the findings of this manuscript could be better highlighted. The interpretations / conclusions presented are mostly focused on the sites described in this study but could also be interpreted in already published papers on this topic.

ANSWER: We believe that the 9 intact sites of the pan-Arctic included in our study are representative enough as they cover main regions of the Arctic. The samples from these sites were taken at the peak of the growing season in summer and processed using the same procedure and are thus comparable. Furthermore, they stem from cryoturbated soils, that were sampled in a highly depth-resolved fashion not involving coring of the active layer.

For methanogens, we showed that their abundances and compositions are site dependent. We believe that adding more comparisons from the other studies will not change the conclusion. For methanotrophs, we found a surprising consistency of high abundance of *Methylobacter*-like phylotypes across the pan-Arctic. We did point out that some other studies also found their dominance in other Arctic regions (L280-286).

Regarding the degraded sites, we added more discussions and interpretation by incorporating published papers. Please refer to **GENERAL ANSWER 2**.

We hope that our explanation and the new contents can minimize your concerns.

*3. I understand why a 16S rRNA approach was only used in this study but was puzzled by why the study did not also look for specific gene markers for methanotrophs (ex. *mmoX*, *pmoA*) and methanogens ex. *mcrA*) which would have supported the 16S results, especially since the 16S sequencing analyses only used a short 250 bp fragments?*

ANSWER: While we generally see the value of a functional marker gene centric approach, either by quantitative PCR or amplicon-based sequencing of either *mcrA* or *pmoA* genes, we believe that it would not have added much extra information to this study that focused on the identity of the members of these functional guilds.

1) Quantitative PCR would have helped to compare the abundance of methanogens and methanotrophs but would not have enabled the taxonomic identification of the genera. Genus identification of all methanogens and most methanotrophic bacteria is well possible via the short

16S rRNA gene fragment derived from a hypervariable region, as applied in this and many other studies.

2) Amplicon sequencing of functional genes would have provided much higher taxonomic resolution but would have prevented the comparison of methanogen and methanotroph abundance in the same sample.

Similarly, metagenomic sequencing of at least some of the samples would have also enhanced, this study, especially, the comparison of the Alaska wet, intermediate, and dry sites?

ANSWER: We agree that metagenomics could also have enhanced the study. However, given the many samples we would have needed to have sound statistical results from metagenomics, this was not possible with the budget given within the project. Next to these budgetary reasons, there are good scientific reasons to not apply metagenomics in our case.

1) MAG-centered metagenomics does not work well for the target groups, due to their very low abundance in the soil microbiome. This would not result in MAGs for methanogens and methanotrophs.

2) Gene centric shotgun analyses would have needed deep metagenomic sequencing to capture enough gene fragment of the target *mcr* and *pmo* genes. Further, the shotgun nature would have prevented in most cases the reassembly of full-length genes that could have been used further for solid taxonomic identification of the methanogen or methanotroph lineages, respectively.

Thus, we believe that the taken methodological approach is of high quality and is appropriate for the identification of key methanogens and methanotrophs in these unique permafrost soil samples from the pan-Arctic.

Minor recommendations

1. If possible please indicate the time of sampling for each of the sampling sites as populations of these microbiomes may have shifted between the winter season and summer season?

ANSWER: The samplings all took place in summer (July or August) at the peak of the growing season. The text was added in the new version.

2. *Some of the terminology used could be more concise please*

ANSWER: We checked through the manuscript and changed the terminology wherever we think might not be concise.

L. 252 *“prevalently”?*

ANSWER: We changed it to “widespread”.

L. 274 *“adjustments” could be changed to adaptations and, if possible, the major adaptations that were found?*

ANSWER: We changed this sentence to the following:

“A recent study showed physiological adaptations of *M. tundripaludum* to varying temperatures by adjusting their central metabolism, protein biosynthesis, cell walls and storage⁶⁸. This might explain the prevalence and dominance of *M. tundripaludum* across heterogeneous Arctic regions.”
(L283-286)

L. 344 *“certain’?”*

ANSWER: This part was removed according to Reviewer 2’s comments.

L. 113-114 *“permafrost sites around the north pole”?* should be changed to Arctic permafrost sites as the “north pole” is not accurate.

ANSWER: Changed.

4. *Figure 1 Panel A is difficult to read and interpret with the different color schemes and the poor quality of the map showing the different sampling sites. What, for example, does the white color indicate? Please also move the Panel Labels (ie A, B, C, etc) to the beginning of Panel description from the end (Please do this for each figure with panels as it makes it easier for the reader?)*

ANSWER: We increased the resolution of Fig. 1 and uploaded the original file of each figure for better resolutions. Additional description regarding white color was added. We also moved the Panel Labels to the beginning of each description for all figures.

Response to Reviewers' Comments

Thank you for taking the time to go through our revisions. We appreciate your positive feedback to our revisions. We have worked to incorporate the two minor suggestions from Reviewer 1.

Reviewer #1 (Remarks to the Author):

This is the revised version of a manuscript I reviewed previously. Overall, the authors have done a great job at addressing all reviewer comments and I think the manuscript can be published in the current form.

I have some minor points:

Fig S1: I think the axis in this figure need to be switched - you want to predict MT/MG RA based on pH etc., not the other way around (so the uncertainty needs to be in terms of MT/MG). This should not affect R values or significance.

ANSWER: We agree with you. We flipped the Fig. S1a-d to have soil parameters on x axis and MG/MT on y axis. Please check the new Fig. S1.

L401-402: "our data suggest that atmMOB might play a pivotal role in increasing the CH₄ sink capacity in drained landscapes after permafrost thaw in a future warmer climate." This sounds like a bit of a tautology to me - isn't CH₄ (net soil CH₄) sink by definition characterized by atmMOB? maybe reword this.

ANSWER: Thank you for finding this tautology. We reworded this sentence as following:

“Although investigations from other Arctic regions are needed for validation, our data suggest that atmMOB in drained landscapes after permafrost thaw might play a more important role in CH₄ dynamics in a future warmer climate.” (L399-401)

Reviewer #2 (Remarks to the Author):

The authors have sufficiently addressed the issues and concerns (e.g., 16S-based taxonomic assignment and various methodological issues, etc.) I raised in the first-round review. I have no

further comments on the revised manuscript.

ANSWER: Thank you for your positive feedback.

Reviewer #3 (Remarks to the Author):

The authors have addressed many of the concerns of this paper appropriately. Some of their responses are somewhat weak, especially the comments on why metagenomic approaches were not used. Nevertheless, I think this paper contains sufficient significant results and novelty.

ANSWER: Thank you for your positive feedback.